# Targeting Human Proteins for Antiviral Drug Discovery and Repurposing Efforts: A Focus on Protein Kinases

**DOI:** 10.3390/v15020568

**Published:** 2023-02-19

**Authors:** Rima Hajjo, Dima A. Sabbah, Osama H. Abusara, Reham Kharmah, Sanaa Bardaweel

**Affiliations:** 1Department of Pharmacy, Faculty of Pharmacy, Al-Zaytoonah University of Jordan, P.O. Box 130, Amman 11733, Jordan; 2Laboratory for Molecular Modeling, Division of Chemical Biology and Medicinal Chemistry, Eshelman School of Pharmacy, The University of North Carlina at Chapel Hill, Chapel Hill, NC 27599, USA; 3Jordan CDC, Amman 11118, Jordan; 4Department of Pharmaceutical Sciences, School of Pharmacy, University of Jordan, Amman 11942, Jordan

**Keywords:** drug discovery intelligence, drug repositioning, host-targeted antivirals immune response pathways, kinases, signal transduction

## Abstract

Despite the great technological and medical advances in fighting viral diseases, new therapies for most of them are still lacking, and existing antivirals suffer from major limitations regarding drug resistance and a limited spectrum of activity. In fact, most approved antivirals are directly acting antiviral (DAA) drugs, which interfere with viral proteins and confer great selectivity towards their viral targets but suffer from resistance and limited spectrum. Nowadays, host-targeted antivirals (HTAs) are on the rise, in the drug discovery and development pipelines, in academia and in the pharmaceutical industry. These drugs target host proteins involved in the virus life cycle and are considered promising alternatives to DAAs due to their broader spectrum and lower potential for resistance. Herein, we discuss an important class of HTAs that modulate signal transduction pathways by targeting host kinases. Kinases are considered key enzymes that control virus-host interactions. We also provide a synopsis of the antiviral drug discovery and development pipeline detailing antiviral kinase targets, drug types, therapeutic classes for repurposed drugs, and top developing organizations. Furthermore, we detail the drug design and repurposing considerations, as well as the limitations and challenges, for kinase-targeted antivirals, including the choice of the binding sites, physicochemical properties, and drug combinations.

## 1. Introduction

Antiviral drugs could be targeted towards either viral proteins (e.g., polymerases, integrase, proteases, accessory proteins, and viral structural proteins), or host proteins that support the viral life cycle [1,2,3]. In 1963, idoxuridine was the first directly acting antiviral (DAA) drug to be approved, initiating a new era in fighting viral infections using drugs that directly target viral proteins [4]. Such drugs gained great attention due to the discovery and licensing of 80 DAA drugs [1]. These drugs were considered safe for humans since they target viral proteins which have no human homologs, except for viral polymerase, which does share some structural similarities with human polymerases, thus being a major reason for nucleoside-based antiviral toxicity [5].

Despite the initial success and low toxicity outcomes on humans, the DAA drug approach faced many hurdles, including the narrow spectrum of activity and the development of antiviral resistance. In fact, every virus has its own set of highly specialized proteins [5]; some proteins share homology among some viruses, but the majority are unique. As a result, broad-spectrum DAA drug discovery efforts often fail and there are only a few other broad-spectrum antiviral drugs in the market that are approved [5]. Therefore, older antivirals often fail to fight novel emerging viruses which threaten the human population by their ability to cause epidemics and pandemics [6]. Additionally, and despite initial success, DAA drugs usually lose their efficacy due to constant viral mutations.

Resistant viral variants emerge over time with different speeds for RNA versus DNA viruses [7]. RNA viruses have mutational rates reaching 10^−4^ (i.e., one mutation per 10,000 base replications), in comparison to mutational rates of 10^−8^ for DNA viruses [8]. Thus, extended exposure to viral infection and continuing viral replication are key factors in the development of antiviral resistance [7]. This led to new legislation for the clinical development of antiviral agents which require testing for viral resistance, mutations, and cross-resistance that persists with treatment [9]. Another hurdle facing the DAA drug approach is the limited number of potential viral proteins that could be targeted with drugs, especially for viruses with small genomes, such as human papillomavirus [7]. 

Targeting human proteins for antiviral drug discovery may result in the development and approval of broader spectrum antiviral drugs that are inherently less susceptible to viral resistance. Thus, host-targeted antivirals (HTAs) are considered promising therapeutic options for combating emerging novel pathogens; even before their genes and proteins are fully characterized [7,10]. Such drugs could potentially lead to universal antiviral agents. Herein, we provide a comprehensive overview of disease-causing viruses, their classification, their life cycle, and their reliance on host kinases to replicate and produce offspring. We also detail the drug discovery strategies that target host kinases to halt the viral life cycle that could potentially lead to universal antiviral agents.

## 2. Disease-Causing Viruses

Disease-causing viruses cause viral infections, which include any illness that is caused by a virus. A virion is the infectious form of the virus that is released from host cells after viral replication [11]. It protects the virus genome and facilitates its entry into specific host cells, which have the required receptors or proteins that facilitate its entry [11,12]. The virion contains the genome, and it is surrounded by a capsid, which is made up of proteins, to protect the genome [13]. Some viruses, such as those from the *Pleolipoviridae* family, do not have capsids [14]. Other viruses have an envelope, which encloses the capsid and is made up of a lipid bilayer embedded with virus-specific glycoproteins, derived from the host’s cell plasma membrane or an intracellular vesicle [13]. Depending on the virus type, other components of virions include mRNAs, proteins and enzymes, and polyamines [13]. 

### 2.1. Classification of Viruses

The International Committee on Taxonomy of Viruses (ICTV) system classifies viruses into different taxonomic levels, starting with realms and ending with species [15,16]. Currently, ICTV’s database of taxonomy shows that there are 10,434 species of viruses [17]. 

There are two different classification systems being adopted for viruses. The classification systems do not correspond to each other, and each is used separately. The first classifies viruses based on their genome; DNA or RNA [13]. They can be viewed in the most recent ICTV Report on Virus Classification and Taxon Nomenclature [18]. The second, which is the Baltimore Classes (BC) system [19,20,21,22], classifies viruses according to the path and process in which the genome is transcribed into an mRNA that is needed for a translation into proteins [13]. The production of mRNA from each type is discussed in the Virus Life Cycle section below. The classification systems are summarized in Table 1. 

### 2.2. Virus Life Cycle

The life cycle of viruses comprises several basic steps starting with viral entry into the host cell and followed by gene expression, gene replication, and finally ending in assembly and viral egress to release new infectious viral particles [13]. However, there could be many differences in how each virus (or class of viruses) achieves these steps. To design effective treatments or preventive therapeutics for viral diseases, especially those targeting host proteins, we need to understand the similarities and differences in the life cycles of viruses [23].

#### 2.2.1. Virus Entry and Uncoating

The entry of viruses into the host cell comprises two stages: attachment and penetration. Entry is later followed by uncoating. These stages would differ between enveloped and non-enveloped viruses. As for enveloped viruses, their membrane would initially fuse with the host cell membrane via the binding of specific viral envelope glycoproteins (also called fusion proteins) to host cell receptors [24]. Consequently, a pore opening (fusion pore) [24,25,26], which might be either enlarged or not [25,27,28,29], allows the passage of the virus core into the host cell cytoplasm [26]. A different mechanism involves the uptake of the virus into an endocytic vesicle followed by the fusion of the viral envelope with the vesicle membrane and releasing the capsid [13]. The fusion mechanisms are either pH-dependent or pH-independent [26]. 

Non-enveloped viruses attach themselves to host cells via either a single protein or multiple protein structures [30,31]. Then, the virus is internalized into the cell by an endocytic vesicle formed via receptor-mediated uptake, followed by the release of the virus and genome into the host cell cytoplasm [13]. Other viruses can inject their genome directly into the host cell cytoplasm across the host cell membrane [13].

Viral attachment to host cells is followed by uncoating, which is the process of releasing the viral genome either into the cytoplasm or directly into the host cell nucleus via the nuclear pores after breaking down viral capsids [13,32]. This step is essential for starting the virus life cycle and permitting the virus to replicate its genetic material.

#### 2.2.2. Gene Expression and Replication

Viral gene expression involves mRNA synthesis (transcription) followed by protein synthesis (translation) inside host cells. Transcription is accomplished via the host and/or viral enzymes, while translation is accomplished via host ribosomes. All viral genomes, irrespective of their type, would be transcribed into mRNA. Gene expression differences do exist between viruses based on the viral genome type. Gene replication, in which a new viral genome is produced to be incorporated into new virions, occurs inside host cells as well. Some viruses undergo replication processes of the genetic material before their genome is eventually transcribed, such as BCII, BCVI, and BCVII viruses [13,22]. On the other hand, BCI, BCIII, and BCV viruses would be transcribed directly, while BCIV viruses are directly translated [13,22].

BCI double-stranded DNA (dsDNA) and BCII single-stranded DNA (ssDNA) viruses, except for poxviruses, have their genome-containing capsids or nucleoprotein complexes moved to the nuclear pores, which permit viral genome entry into the nucleus where gene expression and replication occur [13]. The gene expression of BCIV ((+) ssRNA) viruses starts directly after viral uncoating in the cytoplasm, where viral genomes associate with host ribosomes to start the translation process of viral proteins [13]. However, the genomes of BCIII (dsRNA) and BCV ((−) ssRNA) viruses are associated with the viral RNA-dependent RNA polymerase (RdRp) [13]. Reverse transcribing viruses, BCVI ((+) ssRNA) viruses and BCVII (dsDNA) viruses, associate with the viral reverse transcriptase enzyme (RTz) [13]. A summary of the mechanisms involved in the gene expression and replication of several classes of viruses is presented in Table 2.

#### 2.2.3. Assembly and Egress

The assembly of virions usually takes place at the site of genome replication [13]. Most RNA virions are assembled in the cytoplasm, while most DNA viruses at least start their assembly inside the nucleus. The egress (release) of new virions depends on their type in terms of non-enveloped or enveloped virions [13]. Non-enveloped virions are released after the lysis of the host cell, whereas enveloped virions are released via budding from the host cell and acquiring an envelope from a particular cellular membrane. The envelope could be acquired from the plasma membrane in the final step or could be from the nuclear membrane, Golgi, or other organelle membranes. The new virion is transported via vesicle to the plasma membrane, in which the vesicle fuses with the plasma membrane and release the virion.

## 3. Antiviral Drugs in Different Stages of Clinical Development

According to the Cortellis Drug Discovery Intelligence (CDDI) database [37], there are 41,321 records for drugs in different stages of clinical development, classified as antiviral agents based on “therapeutic group” designation. Among these, 13,027 drugs and biologics target viral proteins (31.5%), while the rest target non-viral proteins [38,39]. The type of drugs, top mechanisms of action, and top targets of antiviral drugs are shown in Figure 1. The top six drug targets are all viral proteins, but host proteins, including indolamine 2,3-dioxygenase 1 (IDO1), toll-like receptor (TLR) 7, programmed cell death 1 (PDCD1), bromodomain containing 4 (BRD4), and a cluster of differentiation 274 (CD274) (also known as Programmed death-ligand 1(PD-L1)) are included. The top non-viral mechanism of action was signal transduction modulation [38]. 

Filtering the complete set of 41,321 antiviral drugs by the following development status criteria: “under active development = yes” and “condition = infection”, resulted in 1013 antiviral drugs (Figure 2), of which 442 drugs are targeting viral proteins (43.6%). The top five infections targeted by these drugs were: coronavirus infections (507 drugs), influenza (162 drugs), herpes (63), respiratory syncytial virus infections (47 drugs), and papillomavirus infections (39).

## 4. Targeting Host Proteins for Antiviral Drug Discovery

Mining the CDDI drug intelligence database indicated that pharmaceutical and biotechnology companies are largely investigating host proteins as targets for antiviral drug discovery efforts [37]. In fact, antiviral drugs targeting viral proteins constituted less than one-third of all antiviral drugs in different stages of drug development. Signal transduction modulators were the most important group of drugs targeting non-viral proteins in the antiviral drug development pipeline. From 41,321 antiviral drug records in different stages of clinical development, there are 3903 signal transduction modulators targeting human proteins, including TLRs (e.g., TLR7, TLR8), kinases (e.g., MAP4K1, CDK9, PIK3, IRAK4, BTK), tumor necrosis factor (TNF), cyclins (e.g., CCNT1), interferons (IFNs), and others. Drugs targeting TLRs and kinases were the most abundant indicating the importance of these two target families in modulating signal transduction in response to viral infections. Figure 3 summarizes signal transduction modulators according to top targets (Figure 3a), top therapeutic groups (Figure 3b), top organizations developing these drugs (Figure 3c), and drug types (Figure 3d).

## 5. Targeting Human Protein Kinases for Antiviral Drug Discovery

Kinase enzymes catalyze the phosphorylation reaction of a broad range of substrates, including lipids, carbohydrates, proteins, and nucleic acids. They are widely present in nature and are involved in a multitude of biological processes [40]. They play key roles in regulating important cellular functions, including metabolism, cell cycle regulation, survival, and differentiation [41]. Considering their vital physiological role, extensive efforts have been directed toward the identification of activity-modulating ligands, such as agonists or antagonists [42]. There are 518 kinases encoded by nearly 2% of the human genome [43]. Interestingly, about half of the kinases are largely uncharacterized with poorly understood roles [40], which may provide a window for the emergence of significant innovational therapies based on the design and development of novel inhibitors of the orphan protein kinases. The recent advancements in science and biotechnology, especially in bioassay design, have enabled the assessment of kinase inhibition in living cells [44], which may offer a valuable tool for shifting from chemical probes to drug candidates. 

Evidently, protein kinases are upregulated in numerous pathological states, and hence their inhibition presents a therapeutic solution for many disorders, including cancer and inflammation [45]. In addition, evidence has begun to accumulate supporting the relationship between kinase inhibition and antiviral activities [46,47]. In fact, host protein kinases are known for their association with the virus life cycle [7,48]. Viruses of both types, DNA or RNA, are known to develop several manipulation mechanisms against the host’s innate and acquired immune mechanisms to control host cellular activities and to create a favorable environment for their replication [48,49,50]. Among these cellular activities is the host’s cell cycle, in which many kinases are involved. Both DNA and RNA viruses manipulate host cell cycle proteins for their own replication advantage. Non-cell cycle kinases also get involved in the progression of viral infections. Most DNA viruses replicate their genetic material inside host cellular nuclei, in the S-phase of the host cell cycle or in arrested cells, such as neurons [48]. Viruses replicating at the S-phase either use the host’s DNA replication proteins that are active at S-phase or require host cellular factors that are activated in the S-phase to express viral proteins required for their DNA replication [48]. Likewise, RNA viruses, replicating in the cytoplasm or nucleus of a host cell, manipulate cell-cycle proteins to replicate their own genetic material [48,50]. RNA viruses activate the IFN-induced double-stranded RNA-dependent protein kinase, known as protein kinase RNA-activated (PKR), which is a member of a small family of serine-threonine kinases that are activated by extracellular stresses [51,52]. The classical activator of PKR is dsRNA, which directly binds PKR and triggers PKR kinase activity [53].

## 6. Kinases as Validated Biomarkers for Viral Infections

To get a better idea about which kinases have any potential for antiviral drug discovery, we mined all drugs and biologics in the CDDI database using the following search criteria: “condition = viral infection” and “mechanism = kinase”. These search criteria allow the retrieval of all drugs and biologics that have been linked to antiviral activity and can also modulate a kinase. Our search resulted in 1251 drugs and biologics, of which 1204 drugs are still in the biological testing phase and have not progressed in the drug development pipeline yet. In fact, 1224 drugs and biologics were coming from patents and could progress in the drug development pipeline soon. The top 10 kinase targets that have drugs “under active development” status for viral infections is protein kinase C (PKC), AXL receptor tyrosine kinase (AXL), Casein kinase II (CK2), mitogen-activated protein kinase (MAPK), mitogen-activated protein kinase kinase (MAP2K; MAPKK; MEK), extracellular signal-regulated kinase (ERK), MAPK p38, cyclin-dependent kinase (CDK) 1 (CDK1), TEK receptor tyrosine kinase, and protein kinase B (PKB; Akt). 

A more targeted search of the CDDI database was performed using the developmental status condition as follows: “development status condition = infection, viral” and “mechanism of action = drugs targeting kinases”. Imposing these filtering criteria ensure that the retrieved drugs and biologics have antiviral effects and are being developed precisely to combat viral infections by targeting kinases. This search resulted in seven drugs and biologics (Table 3). The drug targets of these drugs include casein kinase 2 (CK2), MAP2K, MAPK, MAPK p38, and CDK1. Five of the drugs have the “under active development (UAD)” label, indicating that products are actively moving through the drug research and development (R&D) pipeline from preclinical stages through registration. The launched drug 3-Angeloylingenol is not being investigated for new conditions, and therefore it is not considered UAD. In 2020, this drug was withdrawn from the market in Canada, the European Union, and the United Kingdom.

Mining CDDI for diagnostic and prognostic biomarkers aimed at viral infections resulted in 2994 biomarker records as hits. Among these, 1021 genomic and proteomic biomarkers have reached high validity levels (i.e., were either approved, recommended, or in clinical studies) according to the CDDI database [37]. Kinases make up 4.9% of these 1021 biomarkers, which highlights key roles in the pathogenesis of viral diseases. To get a better idea about the kinase biomarkers for viral infections, we generated a direct protein-protein interactions network using 51 kinases (resulting from the above data-mining effort in CDDI) as nodes (Figure 4). Network edges represented relationships between nodes based on validated experimental protein-protein interaction data extracted from the STRING [54] database. Network generation and visualization were performed in Cytoscape version 3.9.1 [55]. Network generation was followed by a functional enrichment analysis which highlighted the following pathways as top enriched Kyoto Encyclopedia of Genes and Genomes (KEGG) pathways: (1) MAPK signaling pathway; (2) phosphatidylinositol 3-kinases (PI3K)-Akt signaling pathway; (3) Ras-associated protein 1 (Rap1) signaling pathway; (4) RAS signaling pathway; and central carbon metabolism in cancer. The complete enrichment results are provided in Supporting Information (Appendix A).

Notably, the top enriched pathways are directly linked to the virus life cycle and the biological processes that allow the virus to replicate and produce infectious progeny. For example, the MAPK signaling pathway can be activated by a diverse group of viruses [56], and it is involved in the replication of their genetic material. The information in the MAPK pathway is transmitted from one protein to another by phosphorylating serine and threonine residues in a diverse group of proteins leading to a multitude of cellular responses. Furthermore, ERK/MAPK and PI3K/Akt/mTOR signaling responses play a critical role in the pathogenesis of many viral diseases. In fact, viruses such as the Middle East respiratory syndrome coronavirus (MERS-CoV) inhibit these pathways [56], among others. 

Rap1 signaling regulates T-cell and antigen-presenting cell (APC) interactions and modulates T-cell responses to pathogens, such as viruses [57]. In fact, the activation state of Rap1 determines T-cell responses to antigens. Rap1 is also a target for the TLRs, which are key regulators of the immune response to viral infections [58]. Ras signaling was found important for the life cycle of some viruses, including the reovirus. Evidence showed that Ras-transformation affects viral uncoating and disassembly, PKR-induced translational inhibition, generation of viral progeny, the release of progeny, and viral spread [59]. 

It is also known that viruses hijack host metabolic resources and induce a plethora of metabolic alterations in host-cell including host central carbon metabolism. In fact, viral replication relies on extracellular carbon sources, such as glucose and glutamine [60]. The PI3K/Akt/mTOR and HIF-1 signaling pathways regulate glycolysis. Thus, targeting them with inhibitors, such as MK2206 (an Akt inhibitor) or 2-deoxy-D-glucose (2-DG, glycolysis inhibitor), can lower the viral burden in the cells in vitro [61,62,63,64]. 

**Table 3 viruses-15-00568-t003:** Antiviral drugs targeting human kinases in advanced stages of clinical development.

Compound (Route of Administration)	Highest Phase (Condition)	* UAD	Target	Side Effects
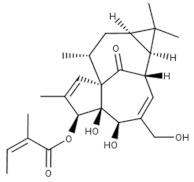 3-Angeloylingenol (Topical)	Launched (Actinic Keratosis)	No	PKC	Local skin reactions at the application site, headache, periorbital edema, nasopharyngitis [1,65]. In 2020, the European Medicines Agency (EMA) recommended the suspension of the product in the EU and EEA as a precautionary measure. Later this year, the marketing authorization was withdrawn by the EMA as the product may increase the risk of skin cancer and the risks outweigh its benefits [37].
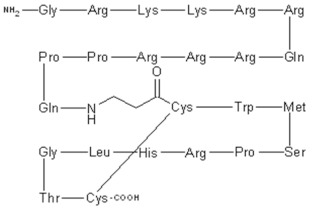 CIGB-300 P15-Tat (Topical)	Phase II (Genital warts)	Yes	CK2	Local adverse events at the injection site including pain, bleeding, hematoma and erythema. Systemic adverse events include: rash, facial edema, itching, hot flashes, and localized cramps [66,67].
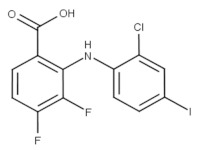 Zapnometinib (Oral)	Phase II/Severe acute respiratory syndrome coronavirus 2 (SARS-CoV-2) infection (COVID-19)	Yes	MEK	Adverse event profile is still unknown. Studies are ongoing [68]. Other MEK inhibitors caused cardiac and ophthalmologic side effects, rash, diarrhea, peripheral edema, fatigue, and dermatitis acneiform [68].
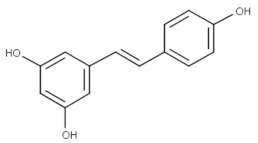 Trans-resveratrol (Topical)	Phase II (Herpes labialis)	Yes	MAPK	Headache, abdominal pain, gastrointestinal problems, urinary tract infections, falls and dizziness [69].
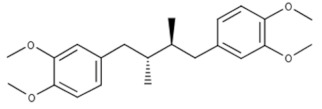 Terameprocol	Phase I (Prevention of HIV transmission)	Yes	CDK1	Ileus, constitutional symptoms, interstitial nephritis, dyspnea and hypoxia, constipation, anorexia [70,71].
POLB-001	Phase I (Influenza)	Yes	MAPK p38	Available clinical data indicated that the drug was tolerated at all tested drug doses with no serious adverse events or trial volunteer withdrawals [10]. It did not elicit liver or cardiac toxicities which could result from the polypharmacological effects of some p38 MAPK inhibitors due to modulating other kinases [72].

* UAD: under active development indicates, indicating that products are actively moving through the drug research and development (R&D) pipeline from preclinical stages through registration.

## 7. The Most Frequently Targeted Kinases for Antiviral Drug Development

### 7.1. Cyclin-Dependent Kinases (CDKs) and Cyclins

More than 20 CDKs have been identified in humans so far. These kinases are involved in cell-cycle progression (e.g., CDK1 to CDK6) and gene expression regulation (e.g., CDK7 to CDK12) [50]. CDKs are controlled by several mechanisms, such as binding to cyclins (ligands), phosphorylation, or binding to CDK inhibitors (CDKis) [50]. CDK inhibitor gene families in mammals include INK4 and Cip/Kip. The INK4 family proteins (p16^INK4a^, p15^INK4b^, p18^INK4c^, and p19^INK4g^) inhibit CDK4/6 activity, whereas the Cip/Kip family members (p21^Cip1^, p27^Kip1^, and p57^Kip2^) modulate the function of several cyclins and CDKs [74,75]. 

DNA and RNA viruses modify CDKs functions to aid their replication by either dysregulating the host’s cell-cycle progression or overcoming restriction factors [76]. These modifications are achieved via specific viral proteins interfering with cyclin/CDKs complexes interactions, phosphorylation of CDKs, or expression of cyclins and CDKis along with other cell cycle proteins, as extensively summarized by Gutierrez-Chamorro et al. [50]. 

It is known that dysregulation of cyclin/CDK functions can lead to uncontrolled cell division and proliferation and thus leading to cancer [77]. Hence, several pharmacological CDKis have been developed and tested clinically for the potential use as cancer therapeutic options [78,79], with only three selective pharmacological CDKis on CDK4/6, namely palbociclib, ribociclib, and abemaciclib, being approved for the treatment of metastatic breast cancer along with hormone therapy by the United States Food and Drug Administration (FDA) and by the European Medicines Agency [50,80]. However, there are currently no approvals for pharmacological CDKis as potential treatment options for viral infections, although there are several promising results in in vitro studies, such as the use of roscovitine, alsterpaullone, dinaciclib, PHA-690509, FIT-039, and abemaciclib [81,82,83,84,85,86]. Moreover, other FDA-approved pharmacological CDKis, such as palbociclib and flavopiridol, were identified as possible treatment options against viruses via simulations or the Library of Integrated Network-Based Cellular Signatures [85,86].

Cyclins could also be targeted to affect the viral life cycle. CCNT1 cyclin T1 gene encodes one of the highly conserved cyclin C proteins. The encoded protein CycT1 is a core subunit of positive-transcription elongation factor b and is highly associated with CDK9. The CycT1-CDK9 complex acts as a cofactor of the HIV-1 Tat protein and is crucial for the amplification of HIV-1 genome RNA.

In addition, this complex is involved in activating transcript elongation via the phosphorylation of Ser2 of the heptad repeats in the carboxy-terminal domain (CTD) of RNA polymerase II (RNApol-II), and subsequently, the phosphorylated RNApol-II initiates the elongation of HIV-1 transcripts [87,88,89,90].

Hence, targeting this family of proteins may interfere with the virus life cycle in terms of RNA synthesis for replication or transcription processes [91]. 

### 7.2. Mitogen-Activated Protein Kinases (MAPKs)

MAPKs are among the cell signaling pathways which are stimulated by extracellular stimuli [92,93,94,95], such as cytokine, hormone, growth factors, pathogens with viruses as an example, osmotic stress, heat, oxidative stress, and microtubule disorganization, resulting in multiple cellular responses [96,97,98]. Among the responses being regulated by MAPKs are gene expression, cell differentiation, cell survival, mitosis, apoptosis, and metabolism [99,100]. The three major MAPK signaling pathways present in mammals are ERK (also known as p42/44 MAPK), Janus kinase (JNK) (also known as stress-activated protein kinase-1 (SAPK1)), and MAPK p38 (also known as SAPK2/RK) [101,102]. RNA and DNA viruses [56] can activate MAPK signaling pathways through their interactions with cell surface receptors known as receptor tyrosine kinases (RTKs), G-protein coupled receptors [95] and integrins [95]. Depending on the type of virus, MAPK pathways could either enhance or hamper virus replication [103] by regulating single or multiple steps in the viral life cycle [104], including viral entry [105,106,107,108], assembly, maturation, release [109,110,111,112], protein synthesis [101], RNA synthesis [108,112], membrane fusion, and cell-to-cell spreading of viruses [111]. In fact, the sustained activation of the MAPK pathway can be achieved by viral secretory proteins, such as the vaccinia virus growth factor (VGF) on the ERK1/2 signaling MAPK pathway [104]. Other effects of MAPK signaling pathways are summarized by Kumar et al. [56].

MAP4K1, also known as hematopoietic progenitor kinase 1, is a member of MAPKs that negatively regulate cellular antiviral responses [112]. Eventually, this will promote the genome replication of viruses and reduce the host’s antiviral responses [112]. Briefly, pathogens, including viruses, can be detected by pattern recognition receptors (PRRs) in a host cell, such as retinoic acid-inducible gene 1 (RIG-I)-like receptors (RLRs) [113] that are a member of the RLR signaling pathway. Subsequently, this will trigger the transcription of downstream regulatory proteins involving TANK-binding kinase 1 (TBK1)/IκB kinase-ε (IKKε), which will mediate the production of IFN-type I (IFN-I) and proinflammatory cytokines to stop the spreading of viruses. MAP4K1 has been shown to promote the degradation of TBK1/IKKε and hence, the inhibition of the cellular antiviral responses. 

As discussed above, the MAPK pathway affects the viral life cycle in different ways, either enhancing or hampering their replication. As potential antiviral agents, there are several kinase inhibitors that target MAPK pathways [56]. However, they are still not approved. Examples of such inhibitors include vemurafenib, U0126, Cl-1040 (PD184352), FR180204, Ag-126, SB203580, SB 202190, AS601245, and SP600125. 

### 7.3. KRR1 Small Subunit Processome Component Homolog

KRR-R motif-containing protein 1 (KRR1) small subunit processome component homolog (hereafter Rev) has been previously known as human immunodeficiency virus 1 (HIV-1) Rev-Binding Protein 2, HRB2, RIP-1, or KRR-R Motif-Containing Protein 1 [114,115]. Rev is one of the HIV-1 viral proteins that are responsible for the nucleocytoplasmic transport of HIV mRNAs as it binds to Rev-response elements (RRE) [116,117,118,119] inside the nucleolus of the host cell [120]. Rev-RRE ribonucleoprotein complexes are then exported to the cytoplasm. This eventually leads to the accumulation of intron-containing unspliced and partially spliced HIV transcripts in the cytoplasm for viral protein expression and viral assembly [121,122]. This nucleolar pathway could be the basis for a novel target for potential treatments against HIV-1 infections [123]. 

### 7.4. Receptor Interacting Protein Kinases (RIPKs)

Receptor-interacting protein kinases (PIPKs), i.e., RIP kinases, are a family of serine/threonine kinases that play a wide variety of functional roles in cellular signaling during pathogen infection. Several RIPKs are involved in necroptosis, which is a form of inflammatory programmed cell death that plays a role in fighting viral infections [124]. Necroptosis leads to cell membrane rupture and leakage of cellular contents [125], thus initiating inflammation and adaptive immunity against viruses [126]. Necroptosis can be achieved through different biological pathways involving different receptors, such as RIPK1, RIPK3, and mixed lineage kinase-like protein (MLKL), which are induced by TNF-alpha (TNF-α) [127,128,129]. Other pathways involve the binding of PRRs, such as DNA-induced activator of IFN (DAI, DLM, or ZBP1), TLRs, and RLRs, directly with RIPK homotypic interaction motif (RHIM) of RIPK3 [130,131,132].

Viruses can inhibit the necroptosis process from facilitating their replication. For example, murine cytomegalovirus and herpes simplex virus-1 (HSV-1) encode viral proteins with inhibitory RHIM that target RIPK1 and ZBP1 [133,134]. *Vaccinia* virus is sensitive to necroptosis mediated by TNF-α and RIPK1 [128,131]. Necroptosis also limits *Vaccinia* virus replication, and RIPK3 deletion in mice caused an increase in virus replication [135]. Furthermore, a study involving the use of swine kidney cell line (PK-15) and pseudorabies virus (PRV) [124] reported the induction of necroptosis by PRV via the RIPK3 pathway. Although PRV has induced necroptosis via the RIPK3 pathway in PK-15 cells, the study has shown that RIPK3 and MLKL knockdown had reduced necroptosis and enhanced PRV replication [124]. Hence, drugs targeting necroptotic pathways to interfere with viral replication and spreading are worth investigating.

### 7.5. Receptor Tyrosine Kinases (RTKs)

#### 7.5.1. Epidermal Growth Factor Receptor (EGFR)

Epidermal Growth Factor Receptor (EGFR) is among the RTKs that are responsible for the regulation of variable cellular processes, such as cell proliferation, differentiation, division, survival, migration, metabolism, and cell cycle control [63,136,137].

Several viruses make use of the EGFR for their own replication and spreading. For example, the *Vaccinia* virus encodes for VGF, which is a homologue of EGF that activates EGFR [138,139,140,141]. EGFR activation is involved in cell motility, a process that is essential for viral infection spreading [142], including the *Vaccinia* virus, as it enhances cell-to-cell contact between infected cells and uninfected cells [143]. Gefitinib, an EGFR tyrosine kinase inhibitor, inhibited the spreading of poxvirus [144]. Hence, targeting EGFR to stop the spreading of the virus is worth further investigation.

Sodium taurocholate cotransporting polypeptide (NTCP) is a transporter expressed in the liver, which is responsible for bile acid uptake, and was found to be involved in hepatitis B virus (HBV) internalization into hepatocytes [145]. HBV internalization is achieved via NTCP oligomerization, a process that is inhibited by troglitazone and NTCP peptide [146,147]. In addition, EFGR interacts with NTCP allowing for HBV internalization [146]. Although gefitinib has no inhibitory effects on NTCP oligomerization, it has interfered with HBV internalization [147]. 

Respiratory viruses, such as influenza virus and Rhinovirus (RV), induce EGFR activation, which its activation inhibits IFN innate antiviral response in airway epithelium [147]. This is achieved via the suppression of IFN regulatory factor (IRF) 1–induced IFN-λ production that is considered the most significant IFN in mucosal antiviral response, and hence, increased viral infection [147].

Moreover, the selective EGFR tyrosine kinase inhibitor, AG 1478, has increased IRF1 and IFN-λ levels and decreased viral titers in vitro and in vivo.

Fucoidan KW (hereafter KW), an agent derived from brown algae *Kjellmaniella crassifolia*, interfered with influenza A virus (IAV) infection in vitro and in vivo [148]. KW interfered with EGFR activation and may inhibit the cellular EGFR pathway as it inhibited IAV endocytosis and EGFR-mediated internalization in IAV-infected cells [148].

EGFR causes airway mucin expression after respiratory syncytial virus (RSV) infections [149]. EGFR tyrosine kinase inhibitors, AG1478 and PD153035, or knocking down EGFR, inhibited the infectivity of RSV A2-2-20F [149].

#### 7.5.2. AXL Receptor Tyrosine Kinase

AXL is a member of the subfamily Tyro-Axl-Mer (TAM) receptor tyrosine kinases. It is one of the phosphatidylserine (PS) receptors present in phagocytes, which also include the T-cell immunoglobulin mucin (TIM) family in addition to TYRO3 and MER tyrosine kinase (MERTK). Generally, PS receptors can clear apoptotic cells, but they may allow viral entry into target cells [147,148,149,150,151,152]. These receptors recognize PS on enveloped viruses and mediate the attachment and internalization of the virus into the target cell via a process called “viral apoptotic mimicry” [83]. The recognition of viral PS is achieved through a PS-binding protein known as growth arrest-specific gene 6 (GAS6) protein, which in turn binds to AXL, and the virus becomes internalized, hence increasing viral infectiousness [153]. For example, AXL promotes the entry of severe acute respiratory syndrome coronavirus 2 (SARS-CoV-2) into human cells [83]. Through in vitro experiments, AXL overexpression enhanced SARS-CoV-2 entry, while its downregulation significantly reduced SARS-CoV-2 infection to pulmonary cells [83]. Hence, targeting AXL receptor kinases may interfere with viral entry and infectivity of other viruses as well.

#### 7.5.3. Janus Kinase–Signal Transducer and Activator of Transcription (JAK-STAT)

##### Proteins

Janus kinase/signal transducer and activator of transcription (JAK-STAT) proteins are intracellular molecules with key roles in signaling pathways modulated by growth factors and cytokines and involved in innate immunity. In fact, innate immunity is the primary defense line for detecting and clearing intruding viruses. Upon viral entry into host cells, viral components will be recognized by PRRs resulting in IFN production. IFNs bind to their respective receptors and activate the JAK-STAT pathway [154], which will result in the production of proinflammatory cytokines and the activation of numerous downstream antiviral IFN-stimulated genes [155]. This process generates an antiviral state that hinders viral replication and provokes the adaptive immune response [155].

The JAK-STAT signaling pathway encompasses three members: tyrosine kinase-related receptors, JAKs, and STATs [156]. Tyrosine kinase-related receptors are transmembrane cytokine receptors that are classified according to the specific cytokine families to which they bind [157]. Recognition and binding of cytokines to their specific receptors result in molecular conformation changes of JAKs (JAK1, JAK2, JAK3, and Tyk2), causing its autophosphorylation or transphosphorylation [158]. Phosphorylated JAKs are activated and may initiate consequent phosphorylation of the receptors and successive docking and phosphorylation of STATs [159]. Phosphorylated STATs form homodimers or heterodimers that are translocated to the nucleus, bind to DNA sequences, and regulate immune-related gene transcription [160,161]. 

Since the activation of the JAK-STAT pathway stimulates the upregulation of immune-related genes against various infections, JAKs were often recognized as potential therapeutic targets. Several JAK inhibitors (JAKis) are permitted for different clinical indications, including the treatment of rheumatologic, dermatologic, hematologic, and gastrointestinal, along with an emergency authorization for Coronavirus disease 2019 (COVID-19) [162]. In addition, ruxolitinib, tofacitinib, baricitinib, and filgocitinib have been recently approved as antiviral agents against HIV, regardless of their original clinical indication [163].

#### 7.5.4. Proto-Oncogene Tyrosine-Protein Kinase MER

The proto-oncogene tyrosine-protein kinase MER, encoded by the *MERTK* gene, belongs to the TAM (TYRO3, AXL, and MERTK) receptor protein tyrosine kinases that regulate several cellular processes, including cell proliferation, cell adhesion, blood clot stabilization, and regulation of inflammatory cytokine release [164]. The TAM receptors are generally expressed by macrophages, dendritic cells, and natural killer cells and are known to antagonize the host’s innate immune responses by a negative feedback loop that hinders the innate immune responses started by TLR and IFN-I signaling pathway [165]. Several reports have demonstrated that the AXL and TYRO3 of the TAM receptors could advance the infection of different viruses in numerous mechanisms [150]. Nonetheless, little is known about the role of MERTK in viral infections. Adomati et al. reported that deletion of MERTK in *MERTK* knockout mice suppressed the release of IL-10 and TGF-β, resulting in the abolition of innate energy, enhanced viral replication, and poor post-infection survival [166]. Recently, Zheng et al. demonstrated that MERTK associates with E2 viral protein to facilitate virus entry. After virus entry, MERTK suppresses mRNA expression of *IFN-β* and promotes viral infection [167]. Hence, this family of kinases could be exploited for the development of novel antiviral drugs.

### 7.6. Cyclin G Associated Kinase (GAK)

Cyclin G-associated kinase (GAK) is a serine/threonine kinase. It plays a major role in clathrin-mediated endocytosis, which is dependent on oligomeric clathrin and adaptor protein complexes [168]. For example, Hepatitis C virus entry and assembly are driven via GAK [169,170]. Other viruses that use this mechanism to infect cells include influenza, dengue, Hantaan virus, Junin arenavirus, HIV, Ebola, and Zika [171,172,173,174]. There are several studies reporting the investigational use of several compounds as antiviral agents via targeting the GAK pathway [170,175,176,177,178].

### 7.7. Leucine Rich Repeat Kinase 2 (LRRK2)

The leucine-rich repeat kinase 2 (LRRK2) protein is a large multidomain protein. It belongs to the ROCO superfamily defined by the presence of tandem Ras of complex (Roc) G-domain and a kinase domain linked by a carboxy-terminal of Roc (COR) sequence. It is expressed in several cells, including immune cells, microglia, and neurons [179,180]. LRRK2 activation is linked to Parkinson’s disease (PD) pathogenesis [181]. Also, HIV-1 infection results in HIV-1-associated neurocognitive disorders (HAND), despite the medication [182,183,184], which shares disease characteristics with other neurodegenerative diseases including PD. HAND is driven by the HIV-1 Tat protein since current medications fail to stop its synthesis [185,186]. Puccini et al. [187] have reported that LRRK2 is a modulator of neuroinflammation and neurotoxicity that results in neuronal damage and causes HAND. LRRK2 kinase inhibitor (LRRK2-IN-1) [188] decreased microglial activation during HAND [189], hence the potential use of LRRK2 inhibitors as antiviral agents is worth further investigation.

### 7.8. Interleukin-1 Receptor-Associated Kinase 4 (IRAK4)

IRAK4 is an important signal transducer downstream of interleukin (IL)-1 receptor (IL-1R), IL-18R, and TLRs. TLRs perform an essential function in triggering the host’s innate immune systems as they take part in the activation of transcription factors and the production of inflammatory cytokines [190].

Upon activation, TLRs commence two major pathways that are essentially reliant on the adaptors: myeloid differentiation primary response 88 (MyD88) and TIR domain-containing adapter-inducing interferon. Myddosome, a bulk oligomeric signaling complex enclosing molecules of MyD88 and members of the IRAK family, forms upon the initiation of MyD88-dependent signal transduction [191,192,193]. Myddosome formation results in IRAK4 autoactivation, which primarily activates IRAK1 followed by IRAK2 [194,195] and eventually activates the TNF receptor-associated factor 6 (TRAF6) [196]. Consequently, TRAF6 triggers the MAPK pathway to activate AP-1 and cAMP-response element–binding protein [197], leading to the production of potent proinflammatory cytokines and chemokines that elicit an acute inflammatory response [197]. Targeting this family of proteins will interfere with the host’s innate immune response toward viral infections.

### 7.9. Transforming Growth Factor-Beta (TGF-β)

TGF-β is one of the highly altered pathways in human cancers affecting cell proliferation, differentiation, apoptosis, and migration [198]. The TGF-β family of cytokines comprises three distinct isoforms, TGF-β1, TGF-β2, and TGF-β3, all of which are dual specificity kinases that bind to the same receptors but with different affinities [199]. TGF-β receptor type 2 (TGFBR2) is a homodimeric receptor that binds to the TGF-β ligand. Upon the ligand binding to the TGFBR2 extracellular domain, a conformational change results in the phosphorylation and activation of TGFBR1, which is a vital propagator of the TGF-β signaling pathway [199]. The Suppressor of Mothers Against Decapentaplegic (SMAD) family of proteins includes secondary effectors of the TGF-signaling pathway [200]. Phosphorylated receptor-specific SMADs associate with SMAD4 and other factors to form a complex that reaches the nucleus and modulates gene expression [201].

Whereas the TGF-β signaling pathway mediates numerous physiological effects, studies have demonstrated that both TGF-β receptors and SMADs crosstalk to other central signaling pathways in the cell, such as MAPK8, MAPK14, PIK3, MAPK1/3 and RAS [200,201]. Some of the downstream mediators of TGF-β signaling are vital cell-cycle checkpoint genes, involving CDKN1A, CDKN1B, and CDKN2B [202,203].

As this family of proteins is involved in gene expression and cell-cycle checkpoints, both of which may be interfered with by viruses for their own replication and transcription, targeting them for antiviral therapy is worth investigating.

### 7.10. Abelson Murine Leukemia (ABL) Viral Oncogene Homolog 1

The ABL family comprises Abelson murine leukemia viral oncogene homolog 1 (ABL1) and Abelson-Related Gene (ARG or ABL2). The encoded proteins have an SH1 catalytic domain and regulatory SH2 and SH3 domains [204]. ABL1 is ubiquitously expressed in the cytosol and nucleus with essential roles in T-cell receptor signaling, cell adhesion and division, gene transcription, and cell growth [204,205].

In HIV infection, ABL1 was shown to modulate actin to tempt cytoskeletal alterations that are crucial for pore formation, pore expansion, and eventually for HIV-1 invading the target T-cell [204]. In addition, during HIV infection, ABL1 was shown to increase tyrosine phosphorylation of RNApol-II, hence promoting the transcription of viral genetic material in invaded T-cells [205]. Hence, targeting ABL1 proteins may interfere with HIV entry and its gene transcription.

### 7.11. Bruton’s Tyrosine Kinase (BTK)

BTK is a TEC kinase with a complex role in n several biological processes including B-cell differentiation. Blocking the expression of BTK demonstrated a reduced maturation of B-cells and decreased serum levels of immunoglobulins in mice [206]. In addition, BTK takes part in different signaling pathways that modulate both innate and adaptive immune responses [207]. Several studies have investigated the key role of BTK in viral infections [208]. After the recognition of viral ssRNA in macrophages, TLRs initiate signaling through BTK-dependent activation of NF-κB leading to increased production of various inflammatory cytokines and chemokines, in addition to phagocytosis [208]. BTK would form another target to investigate novel antiviral drugs as well.

## 8. Kinases Inhibitors in Various Developmental Stages as Antiviral Drugs

In this section we provide an overview of important kinase inhibitors in various developmental stages as antiviral drugs. These inhibitors are listed in Table 4 and are given numbers from 1 to 45.

### 8.1. Mitogen-Activated Protein Kinases Inhibitors

Biological data showed that U0126 (1), a MEK1/2 inhibitor, exerts an 80% reduction in virus titers in Madin-Darby canine kidney (MDCK) cells at 50 μM and displays low values of a multiplicity of infection; a value of 1 and 0.0025 after 9 and 48 h, respectively [209,210,211]. Moreover, **1** exhibits antiviral activity against mutant ERK. Studies showed that if **1** does not impede viral RNA or protein synthesis, the viral ribonucleoprotein (RNP) sequesters in the nucleus and consequently suppresses viral production [211]; **1** inhibits influenza B virus (IBV) and IAV growth in MDCK cells, avoiding the evolution of resistant strains such as the H1N1 outbreak [210]. Further studies reveal that **1** inhibits astrovirus proliferation in human colorectal adenocarcinoma (Caco-2) cells and suppresses all viral life cycle stages [212]. Recent studies reported that **1** blocks murine coronavirus proliferation [213]. 

Biological investigation declared that SP600125 (**2**), a JNK-1 inhibitor, demonstrates a promising effect against Japanese encephalitis virus (JEV) [46,214,215,216]. Studies declared that **2** decreases the inflammatory cytokines generated by JEV-infected microglia (BV-2) and glioma (U251) cell lines [217]. Animal studies reported that **2** reduces the viral titers in infected mice brains and increases the survival rate [46]. Further studies declared that **2** impedes the growth of SARS-CoV in infected Vero E6 (African green monkey kidney epithelial) cells [218].

A kinome analysis experiment declared that SB203580 (**3**), a p38 inhibitor, and **1** exert an antiviral activity by 45% and 51% at 10 μM, respectively [219,220,221]. Furthermore, **1** exerts more antiviral activity in preinfected cells compared to its inoculation after 2 hours of infection [219]. Biological studies announced that **3** impaired the phosphorylation of heat shock protein 27 (HSP27), cAMP response element-binding protein (CREB), and eukaryotic translation initiation factor 4E (eIF4E) in SARS-CoV-2 infected cells [220]. 

A repurposing study revealed that the approved drugs selumetinib (**4**) and trametinib (**5**), MAPK inhibitors, exert a superior antiproliferative activity (>95%) against MERS-CoV [221,222,223]. Recent studies declared that **4** and **5** induce immune cells, decrease angiotensin-converting enzyme 2 (ACE2) encoding in human cells, and decrease cytokine expression in COVID-19 patients’ blood [224,225,226]. Such findings interrogate that **4** and **5** mitigate SARS-CoV-2 infection and control the body’s immune reactions [224,225,227]. Another repurposing study declared that papaverine (**6**), an alkaloid, inhibited both IAV and paramyxoviruses with an IC_50s_ range of 2.0-36.4 μM [228,229]. Biological data reported that **6** inhibits MEK and ERK phosphorylation and sequentially impedes the release of viral RNP, while it does not affect viral RNA synthesis [230]. Data propose that **6** exhibits its inhibitory activity at the terminal stage of the viral life cycle [229,230]. Recent studies revealed that **6** suppresses the cytopathogenic effects of SARS-CoV-2 (EC_50_ = 1.1 μM) [231].

ATR-002 (**7**), a MEK inhibitor released by Atriva Therapeutics in Tübingen, Germany, shows antiviral activity [232,233,234,235]. Compound **7** is the active metabolite of CI-1040 that was disused due to low plasma concentration, and the development of CI-1040 was ceased despite the fact that it was shown to exert a better MEK inhibitory activity than that of CI-1040 with an IC_50_ value of 5.3 nM in contrast to 17 nM [236]. Studies claimed that **7** exerts a wide spectrum of antiviral activity in vitro and in vivo mouse models against diverse influenza virus strains of IAV and IBV [232]. In addition, studies reported that **7** exerts a safe, selective, and effective anti-SARS-CoV-2 activity [233,234]. Clinical studies showed that **7** terminates Phase I trials successfully in 70 healthy volunteers (Clinical Trial Identifier: NCT04385420) [31]. Compound **7** is undergoing a Phase II trial to probe its activity against COVID-19 (EU Clinical Trial Register Number: 2020-004206-59) [32].

### 8.2. Cyclin-Dependent Kinases Inhibitors

Palbociclib (Ibrance^®^) (**8**) [237,238], ribociclib (Kisqali^®^) (Ribotix^®^) (**9**) [239,240,241], and abemaciclib (Verzenio^®^) (**10**) [242] are FDA-approved CDK4/CDK6 inhibitors that are used therapeutically for breast cancer [37,243]. Biological studies showed that **8** suppresses HIV-1 (EC_50_ = 0.016 μM) and HSV-1 (EC_50_ = 0.020 μM) proliferation in vitro [244]. Studies declared that abemaciclib (**10**) exerts an in vitro inhibitory activity against SARS-CoV-2 with CC_50_ > 50 mM and IC_50_ = 6.6 mM [84]. Another study reported that **8** impairs HIV-1 reverse transcription and proliferation as well, as **8** decreases the activation of sterile α motif and HD domain-containing protein-1 (SAMHD1) in CD^4+^ T-lymphocytes, macrophages [245] and HSV-1 [244]. A recent study declared that CDK4/CDK6 inhibitors (**8**, **9**, and **10**) might impede SARS-CoV-2 infection in breast cancer females [241].

Studies declared that (*R*)-roscovitine (**11**), a purine-based inhibitor, disrupts DNA synthesis of human cytomegalovirus (HCMV) [82]. A further study reported that **11** exhibits antiproliferative activity against the human Polyomavirus JC virus (JCV) by impairing the transcription of JCV late genes suggesting that CDKs are required for JCV DNA transcription and replication [80]. Flavopiridol (**12**), a CDK1/CDK2/CDK4 inhibitor, suppresses HIV-1 proliferation by disrupting Tat-transcription [246,247,248]. A high-throughput screening revealed that **12** inhibits multiple IAV strains in human lung adenocarcinoma (A549) cells without cytotoxicity [83]. Recent reports proposed that **11** and **12** could be new therapeutic agents against COVID-19 based on their wide-spectrum antiviral activity and safety window as anticancer agents. 

Biological screening studies declared that alsterpaullone (**13**), CDK1/CDK2 inhibitor, a purine analogue, exerts potent activity against HIV-1 [80,249]. Studies reported that **13** exerts a selective and dose-dependent antiproliferative activity in an infected chronic HIV-1 infection model (ACH2), acute HIV-1 infection model (OM10.1), latent HIV-1 infected T-lymphocytic model (J1.1) cells [80]. Studies reported that **13** exerts 100% inhibition against CDK2 in an infected cell at 0.5 mΜ [80]. Also, studies revealed that CDK2 expression and protein levels are suppressed in infected HIV-1 cells following **13** inoculations. And, studies reported that **11** and **13** demonstrate a synergism effect in infected human peripheral blood mononuclear cells confirming that targeting numerous CDKs might be promising to eradicate HIV-1 [80]. Studies reported that **13** might exert anti-SARS-CoV-2 activity [81].

A screening study disclosed FIT-039 (**14**) as a CDK9 competitive inhibitor (IC_50_ = 5.8 μM) [250]. Data showed that **14** exerts dose-dependent inhibitory activity against HSV-1 in infected cells [250]. Furthermore, **14** attenuates the phosphorylation of RNApol-II CTD in HSV-1 infected human embryonic kidney (HEK293) cells, hypothesizing that **14** might impede viral transcription. The biological investigation reported that **14** ameliorates skin injury and survival rates in mice models with HSV-1 infected skin. Data showed that **14** exerts antiproliferative activity against human adenovirus (HAdV) type 5, HSV-2, HBV, and HCMV [83]. Studies announced that dinaciclib (**15**), a CDK1/CDK2/CDK5/CDK9 inhibitor [251], exerts antiviral activity against SARS-CoV-2 [252]. Another investigation declared that a combination of **12** and **15** induces a synergistic effect [83]. Biological studies revealed that PHA-690509 (**16**), a CDK2 inhibitor, prevents Zika virus (ZIKV) infection in human glioblastoma (SNB-19) cells in a dose-dependent manner (IC_50_ = 1.72 μM) by inhibiting RNA replication [81]. 

### 8.3. Receptor Tyrosine Kinase Inhibitors

Studies reported that genistein (**17**), RTK inhibitor, and gefitinib (**18**), narrow spectrum EGFR inhibitor, impede IAV entry in A549 cells [253]. Additional studies showed that **17** inhibits the proliferation of HIV-1 [254], HSV-1 [255], arenavirus [147], and SARS-CoV [256,257]. Computational studies disclosed that **17** might be a possible SARS-CoV-2 inhibitor [258]. Biological studies showed that **18** impede SARS-CoV-2 proliferation in Caco-2 and Vero E6 cells [259]. Further studies declared that AG1478 (**19**), selective EGFR, hinders transmissible gastroenteritis virus (TGEV) uptake by porcine intestinal columnar epithelial cells (IPEC) [147]. However, studies showed that **19** exhibits adverse effects in rhinovirus and IAV-infected human bronchial epithelial (BEAS-2b) cells [147]. A recent hypothesis stated that EGFR inhibitors exemplified by gefitinib (**18**) and erlotinib (**20**), as well as platelet-derived growth factor receptor (PDGFR) inhibitors such as sorafenib (**21**), impede the development of lung fibrosis [112,260]. Though, the advantageous effect of such kinase inhibitors is not related to antiviral activity [149]. Studies revealed that **21** inhibits HBV gene expression by downregulating the JNK cascade, which forms FXR; a transcription protein that incites HBV growth and gene expression [261]. Further studies showed that **21** suppresses the proliferation of SARS-CoV-2 and DNA viruses in vitro [262,263]. Nintedanib (**22**), a fibroblast growth factor receptor (FGFR) inhibitor, reaches clinical trials as a treatment for lung fibrosis for moderate symptoms of COVID-19 [70].

Studies disclosed that R428 (**23**) impairs the kinase activity of the AXL receptor, a member of the RTKs family, and consequently inhibits viral entry and enhances IFN-1 signaling transduction [54]. Recent studies reported that suppressing the AXL receptor attenuates the viral infection in human lung adenocarcinoma (H1299) and lung epithelial cells [264]. Studies reported that AG879 (**24**), tropomyosin receptor kinase-A (TRK-A) and EGFR/HER2 inhibitor, and tyrphostin A9 (**25**), PDGFR inhibitor, block the replication of IAV [110]. Another study declared that **25** impedes TGEV replication in PK-15 or ST cells [55].

### 8.4. Numb-Associated Kinase Inhibitors

The human Numb-associated kinase family of Ser/Thr kinases harbors adaptor-associated kinase 1 (AAK1), BMP-2 inducible kinase (BMP2K/BIKE), GAK, and Ser/Thr kinase 16 (STK16) [265]. Biological data recorded that erlotinib (**20**) and sunitinib (**26**), AAK1 and GAK inhibitors, exert a wide-spectrum antiviral activity against RNA viruses descending from *Arenaviridae, Flaviviridae, Filoviridae, Paramyxoviridae,* and *Togaviridae* [108,266]. Another study revealed that **26** attenuates SARS-CoV-2, SARS-CoV, and MERS-CoV infection by blocking the adaptor-related protein complex 2 subunit mu 1 (AP2M1) phosphorylation [267]. Biological studies suggested that baricitinib (**27**), an effective AAK1 and GAK inhibitor, might be a potent therapy for COVID-19 [265,268,269]. Clinical studies reported that **27** reduces ICU admission and ameliorates signs and symptoms of COVID-19 patients [270]. Another study showed that **27** blocks of Janus kinase (JNK1/2) anticipate potential risks to take place [271]. Therefore, treatment with **27** is recommended for short intervals (7–14 days) to bypass opportunistic viral infections [73]. 

### 8.5. Src Kinases Inhibitors

Studies reported that saracatinib (**29**) exerts antiproliferative activity against MERS-CoV (EC_50_ = 2.9 μM and CC_50_ = 50 μM), human coronaviruses (HCoV-229E, EC_50_ = 2.4 μM) and (HCoV-OC43, EC_50_ = 5.1 μM) [204]. Further studies revealed a strong correlation between Src kinase suppression and Dengue virus (DENV) inhibitory profile [108]. Studies reported that **28** and **29** decreased DENV infection in Huh7.5.1, clone C6/36 (derived from Aedes albopictus mosquito), and Vero (derived from an African green monkey kidney epithelial) cells in a dose-dependent manner starting from 0.05–5 μM. Recent studies showed that **28** and **29** suppressed RNA replication of DENV through Fyn kinase [272]. Studies declared that bosutinib (**30**), an Src kinase inhibitor, inhibits SARS-CoV-2 entry with (EC_50_ = 2.45 µM) [273,274,275]. Studies reported that ponatinib (**31**) suppresses cytokines release responding to the SARS-CoV-2 N-terminal domain. Also, studies showed that **31** inhibits in vitro cytokine release in myeloid cells and in vivo in lung inflammation mouse models [276]. Biological investigations disclosed that vandetanib (**32**) inhibits SARS-CoV-2 in A549 cells expressing ACE2 (A549-hACE2) with an IC_50_ value of 0.79 μM as well as it suppresses HCoV-229E proliferation [277]. In addition, studies proposed that **32** impede COVID-19 cytokine storm in mice models [277]. The in vivo findings revealed that **32** decreases IL-6 and 10, TNF-α, and inflammatory infiltrate in infected mice lungs without decreasing viral titers [277]. 

Studies reported that dasatinib (**28**), an Src kinase inhibitor, impedes HIV-1 infection through the entry process at the membrane hemifusion stage and targets the viral restriction factor SAMHD1 [278]. Biological studies showed that inoculation of glioblastoma (U87/CD4/CCR5) cells with 300 nM of **28** reduces cellular fusion with HIV-1 by 93% [204]. Studies showed that a combination of **28** with sofosbuvir, a viral entry inhibitor, exhibits a synergistic activity against HCV. Such synergism effect potentiates HCV antiviral activity and improves IC_50_ values by 210-fold in human hepatoma (Huh7.5.1) cells [110]. Studies demonstrated that **28** exhibit antiviral activity against MERS-CoV and SARS-CoV [279,280]. 

### 8.6. Phosphatidylinositol-3-Phosphate-5 Kinase Inhibitors

Studies declared that apilimod (**33**), YM-201636 (**34**), and vacuolin-1 (**35**), phosphatidylinositol-3 phosphate-5 kinase (Pikfyve) inhibitors, inhibit filoviruses and Ebola virus (EBOV) [281], disturb the endosomal trafficking, and consequently block SARS-CoV-2 entry [282]. Further studies showed that **33** exerts antiviral activity against the SARS-CoV-2 and Omicron variants [283]. Compound **33** has reached Phase II clinical trials (NCT04446377) to probe its effect on the progress of COVID-19 [108].

### 8.7. G-Protein Coupled Receptor Kinase Inhibitors

Studies showed that methyl 5-[2-(5-nitro-2-furyl) vinyl]-2-furoate (**36**), G-protein coupled receptor kinase (GRK) inhibitor, blocks IAV entry and replication [47].

### 8.8. Abelson Tyrosine Kinase Inhibitors

Drug repurposing studies identified imatinib (**37**) and **28**, ABL inhibitors, as anti- MERS-CoV and SARS-CoV agents, whereas nilotinib (**38**) exerts anti-SARS-CoV activity [280]. Recent studies showed that **35** inhibits SARS-CoV in Vero E6 (EC50 = 9.82 μM) [284]. Furthermore, **37** has reached Phase III clinical trials for COVID-19 (NCT04394416) [114]. Another study displayed that GNF-2 (**39**), an ABL inhibitor (IC_50_ = 138 nM), exerts anti-DENV activity by inhibiting viral entry and proliferation [285]. Studies confirmed that **39** targets both ABL (intracellular) and DENV E protein (extracellular). Data showed that **39** exerts efficient inhibitory activity against DENV E protein (IC_50_ = 5−25 μM) [285]. Further studies demonstrated that **37**, **39**, and GNF-5 (**40**) impede the fusion of the S protein of MERS-CoV and SARS-CoV and consequently block coronavirus entry [286,287]. 

### 8.9. Calcium/Calmodulin-Dependent Protein Kinase II Inhibitors

Studies showed that BSA9 (**41**), calcium/calmodulin-dependent protein kinase II (CaMKII) inhibitor (IC_50_ = 0.79 μM), exerts anti-DENV (EC_50_ = 1.52 μM) and anti- ZIKV (EC_50_ = 1.91 μM) in neuroblastoma (BE (2)-C) cells through inhibiting viral entry [288,289,290].

### 8.10. Ataxia Telangiectasia and Rad3-Related Kinase Inhibitors

Studies showed that berzosertib (**42**), ataxia telangiectasia, and rad3-related kinase (ATR) inhibitors exhibit antiviral activity in cells overexpressing ACE2, such as human embryonic kidney (HEK293T) and Vero-E6 cells [291]. ATR inhibitors causing DNA damage are suggested as a new therapeutic agent for SARS-CoV-2 [292]. Another study declared that **42** exerts anti-SARS-CoV-2 and MERS-CoV activity in an array of cell lines [292]. 

### 8.11. Adenosine 5′-Monophosphate-Activated Protein Kinase Inhibitors

Studies revealed that dorsomorphin (**43**), an adenosine 5′-monophosphate-activated protein kinase (AMPK) inhibitor, impairs EBOV proliferation in a dose-dependent manner in vitro in Vero cells and mitigates human macrophages infection that is EBOV targets in vivo [293,294,295]. Studies spotlighted the significance of **43** as anti-SARS-CoV-2[296].

### 8.12. Phosphatidylinositol 3-Kinase/Akt/mTOR Pathway Inhibitors

Studies showed that LY294002 (**44**), a PI3K inhibitor, impedes the entry of Zaire EBOV into Vero-E6 cells [297]. Further studies reported that **44** impairs the entry of HSV-1 [298] and African swine fever virus [299]. 

### 8.13. Sphingosine Kinase Inhibitors

Studies displayed that sphingosine kinase (SphK) inhibitors decrease SARS-CoV-2 replication and viral titer [300]. Studies showed that opaganib (**45**), SphK2 inhibitor (Ki = 9.8 μM) [301], mitigates inflammatory signaling and decreases viral growth in vitro in human bronchial tissue [302]. Compound **45** has reached Phase II/III clinical studies for COVID-19 pneumonia patients (NCT04467840) treatment [119]. 

## 9. Repurposing Old Kinase Inhibitors as Antivirals

Currently, most approved kinase inhibitors are clinically used for the treatment of cancer and inflammatory processes [38,303,304]. However, only 50 of the 500 known protein kinases encoded by the human genome have been targeted by drugs for cancer treatments [305,306,307]. The available kinase inhibitors provide a unique opportunity to expand therapeutic avenues beyond their current uses. Viral infections stand as a global issue challenging the health system and demanding successful therapeutic strategies to tackle this issue. Viruses are dependent on several host cellular proteins, and phosphorylation events appear to have a crucial role during viral progression at distinct stages of infection [221,308,309,310].

Repurposing the existing kinase inhibitors to combat viral infections may provide a better understanding of the kinases’ roles during a viral infection. In addition, since clinical data on toxicity, pharmacokinetics, and pharmacodynamics are already available for the approved inhibitors, repurposing them for a new clinical application would shorten the clinical pathway and limit the cost of their release as approved drugs. 

## 10. Pros and Cons of Targeting Host Proteins for Antiviral Drug Discovery

Targeting host proteins via the use of HTA drugs is considered a new approach to the treatment of viral infections. In contrast to DAA drugs, HTA drugs have been shown to have broad spectrum of activity and low susceptibility to viral resistance [5,311]. This may be justified as host-cellular factors and/or mechanisms are used by viruses for their replication and transcription, as discussed earlier in this review, irrespective of their type, hence increasing the antiviral spectrum. In addition, viral replication errors, especially RNA viruses, contribute to resistance and negatively affect DAAs’ activity [5]. As host proteins are not subjected to viral genetic control and their resistance-contributing replication errors, along with low host genetic variability compared to viruses, HTA drugs are less likely to be ineffective against viruses [5,311]. 

On the other hand, HTA drugs have some disadvantages. HTA drugs target host cell cellular pathways required for survival, thus contributing to adverse effects and cytotoxicity [312,313]. In phase II clinical trials, alisporivir has been shown to cause hyperbilirubinemia and hypertriglyceridemia [312,314,315]. The use of HTA drugs may also cause the viruses to use alternate hosts or modify their affinity towards them [311]. Moreover, genetic polymorphism among patients may contribute to the reduced efficacy of HTA drugs to block their targets [314]. For example, suboptimal response, using alisoprivir against HCV, was observed in 10–15% of patients in the study [311]. Furthermore, poor correlation between in vitro, in vivo, or clinical trials responses to HTA drugs were observed, such as the use of VX-497 and statins for the treatment of HCV infection [316,317,318].

In relation to kinases, an in vitro study has investigated the use of specific kinase inhibitors targeting MEK and Src kinases to evaluate their activity against the proliferation of flavivirus infections inside BHK21 and Vero cells [221]. BHK21 and Vero cells are mammalian cells usually used to study the growth of viruses in laboratories [308,309]. The MEK inhibitors (trametinib and selumetinib) and Src inhibitors (saracatinib and bosutinib), which are being designed to treat cancers, showed antiviral activity against several flaviviruses (Zika virus, dengue virus, and yellow fever virus) [221]. However, the most effective and safest among them was trametinib. Safety was evaluated via the calculation of the selectivity index (SI), in which a ratio between the 50% cytotoxic concentration (inhibitors against cells only) and the 50% effective concentration (antiviral activity) is calculated. Trametinib had the highest SI compared to the other kinase inhibitorsv [221]. Hence, there is a risk of cytotoxicity when repurposing kinase inhibitors to target viral proliferation.

RV replication depends on phosphatidylinositol 4-kinase III beta (PI4KIIIβ) [310]. In in vitro and in vivo studies, the replication of RV has been prevented by the use of aminothiazole compounds, Compound 1 and Compound 2, which are PI4KIIIβ inhibitors [310]. However, the in vivo study has shown several adverse effects on mice, including muscle weakness and difficulty in breathing [310]. Although it was concluded that these adverse effects could possibly be species-specific and not relevant to humans, further in vivo studies are required to check the mechanism of toxicities and their relevance to humans [310].

Kinase inhibitors were also recently being investigated clinically in patients with COVID-19, as extensively summarized by Malekinejad et al. [319]. Clinical trials have studied, with some still ongoing, the use of JAK/STAT and BTK inhibitors for patients with COVID-19. Although a number of JAK/STAT kinase inhibitors, such as NCT05187793 and NCT04390061, resulted in clinical recovery and prevention of severe respiratory failure, respectively, others have shown adverse outcomes [319]. Among the adverse outcomes resulted due to the use of JAK/STAT and BTK inhibitors include hospitalization, the need for mechanical ventilation, extracorporeal membrane oxygenation (ECMO), respiratory failure, the need for supplemental oxygen, renal failure, disease progression, ICU admission, and death [319]. 

HTA drugs that target kinases, in particular, may halt viral life cycles and be curative if we are successful in detecting viral infections early on. Although most anticancer kinase inhibitors are not curative as tumors find escape routes, this may or may not be the case if they were to be used for the treatment of viral infections. Resistance to kinase inhibitors in cancer patients could be either primary (innate) or acquired. Innate resistance is caused by tumors that harbor specific genetic mutations, which cause cancer cells to become refractory to target inhibition. Acquired resistance, on the other hand, occurs due to complex, diverse mechanisms, including secondary target mutations that result in the acquisition of ‘bypass’ signaling pathways, the alteration of the cancer microenvironment, and increased expression of drug transporters, such as multidrug resistance proteins or brain cancer resistance proteins. These mechanisms, if they occur in infected host cells, may prevent antiviral kinase-inhibiting drugs from reaching clinically effective concentrations inside infected cells.

Targeted therapies that involve the use of enzyme inhibitors, such as kinase inhibitors, can alter normal physiological cell functions, thus altering normal cell behavioral phenotype via interacting with other pathways that are not intended to act upon [320,321,322]. These abnormal effects could arise either directly due to off-target interactions with additional proteins within the targeted pathway [323,324] or indirectly due to crosstalk between the targeted pathway and other pathways regulating a behavioral response [325]. In addition, off-target effects could arise due to retroactivity, in which a downstream disruption in a signaling pathway produces an upstream effect even though there is no association of negative feedback inhibition [326]. Although retroactivity does occur naturally in covalently modified cascades, the movement of signals within signaling pathways is usually in a downstream manner [327]. Kinase inhibitors have been shown to produce off-target effects via retroactivity [327]. Cell behaviors and off-target effects in the presence of kinase inhibitors or other drugs could be predicted computationally using a partial least square regression framework based on the signals of several key signaling pathways [328,329,330].

A summary of the pros and cons of targeting viral proteins or host proteins is presented in Table 5.

## 11. Drug Design Considerations for Kinase-Targeted Antivirals

### 11.1. Binding Sites

Kinase inhibitors can be classified into six main groups according to their binding sites (Table 6): (1) inhibitors that bind to the ATP pocket in the active conformation of a kinase and are known as type I inhibitors; (2) inhibitors that bind adjacent to the ATP pocket (adenine binding residues) of the unphosphorylated inactive conformation of kinases and are knowns as type II inhibitors; (3) non-ATP competitive inhibitors that bind within the cleft between the small and large lobes close to the ATP binding pocket and are known as type III inhibitors; (4) allosteric inhibitors that bind away from the ATP cleft and are knowns as type IV inhibitors; (5) agents that span two distinct regions of the protein kinase domain and are known as type V inhibitors; and (6) agents that form covalent bonds with their target kinases and are known as type VI inhibitors.

### 11.2. Physicochemical Properties

One feature of kinase inhibitors that have transformed clinical cancer care is their ability to penetrate the blood–brain barrier [305]. This could be advantageous for antivirals since the drug needs to overcome drug delivery barriers to reach its intracellular or intranuclear targets. 

### 11.3. Kinase Inhibitor Combinations

Designing drug combinations of kinase inhibitors, as well as combinations of kinase inhibitors with other therapeutic modalities, holds the promise of identifying effective antiviral drugs by targeting signal transduction pathways at multiple points, which could impede the virus life cycle faster, in addition to better modulation of the immune response and inflammatory pathways that are turned on or off in response to viral infections.

## 12. Conclusions

HTAs have vast potential to treat or prevent viral infections in addition to combating emerging novel viruses. They are conceivable to develop universal antivirals with an increased antiviral spectrum and reduced resistance. Certainly, viruses exploit host proteins to enter host cells, replicate their genomes, synthesize their own viral proteins, and produce a progeny of infectious viral particles. This review provided a summary of host proteins involved in the life cycle of viruses and provided an important update on the antiviral drug development pipeline with a special focus on kinase-targeting antivirals detailing their role in signal transduction pathways and providing drug discovery intelligence on drugs at different developmental stages. We also discussed repurposing approved kinase inhibitors, which have been clinically used for the treatment of cancers and inflammation, to combat viral diseases. However, more progress is needed to ensure that R&D efforts continue to identify novel kinase ligands or repurpose already existing kinase-targeting drugs to treat or prevent existing and emerging viral diseases. Indeed, the potential for developing novel antiviral kinase inhibitors is massive and will continue to be a major growth area for modulating infectious and auto-immune diseases in the future.

## Figures and Tables

**Figure 1 viruses-15-00568-f001:**
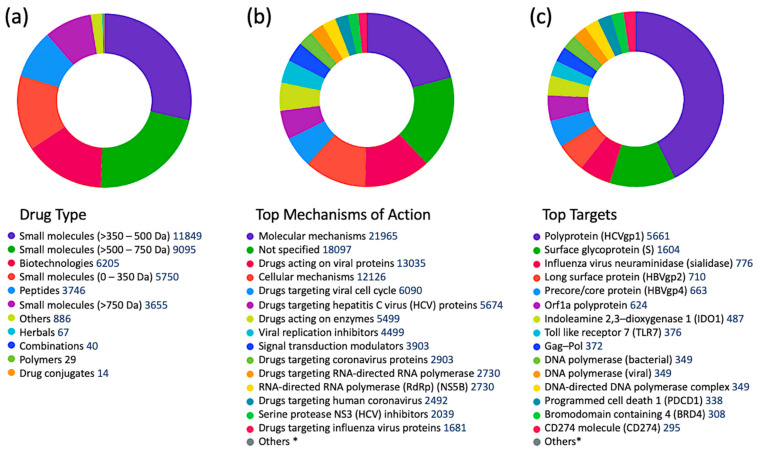
An overview of antiviral drugs according to (**a**) drug type, (**b**) top mechanisms of action, and (**c**) top targets. * Others in b and c indicate that there are other mechanisms and targets that are not shown in the figure because they represent a large number of categories with very small percentages. Data source: Cortellis Drug Discovery Intelligence [37], https://www.cortellis.com/drugdiscovery/, accessed on 23 December 2022, ©2022 Clarivate. All rights reserved.

**Figure 2 viruses-15-00568-f002:**
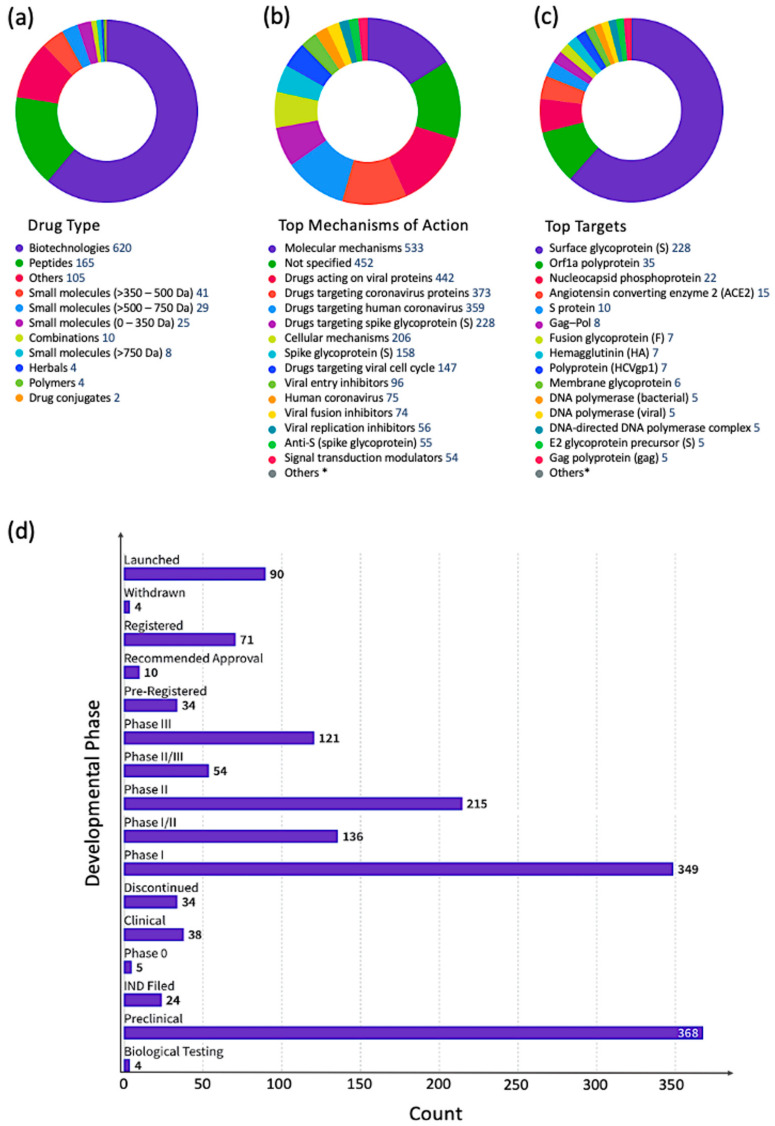
An overview of antiviral drugs under active development for viral infections according to (**a**) drug type, (**b**) top mechanisms of action, (**c**) top targets, and (**d**) developmental status. * Others in (**b**,**c**) indicate that there are other mechanisms and targets not shown in the figure because they represent a large number of categories with tiny percentages. Data source: Cortellis Drug Discovery Intelligence [37], https://www.cortellis.com/drugdiscovery/, accessed on 24 December 2022, ©2022 Clarivate. All rights reserved.

**Figure 3 viruses-15-00568-f003:**
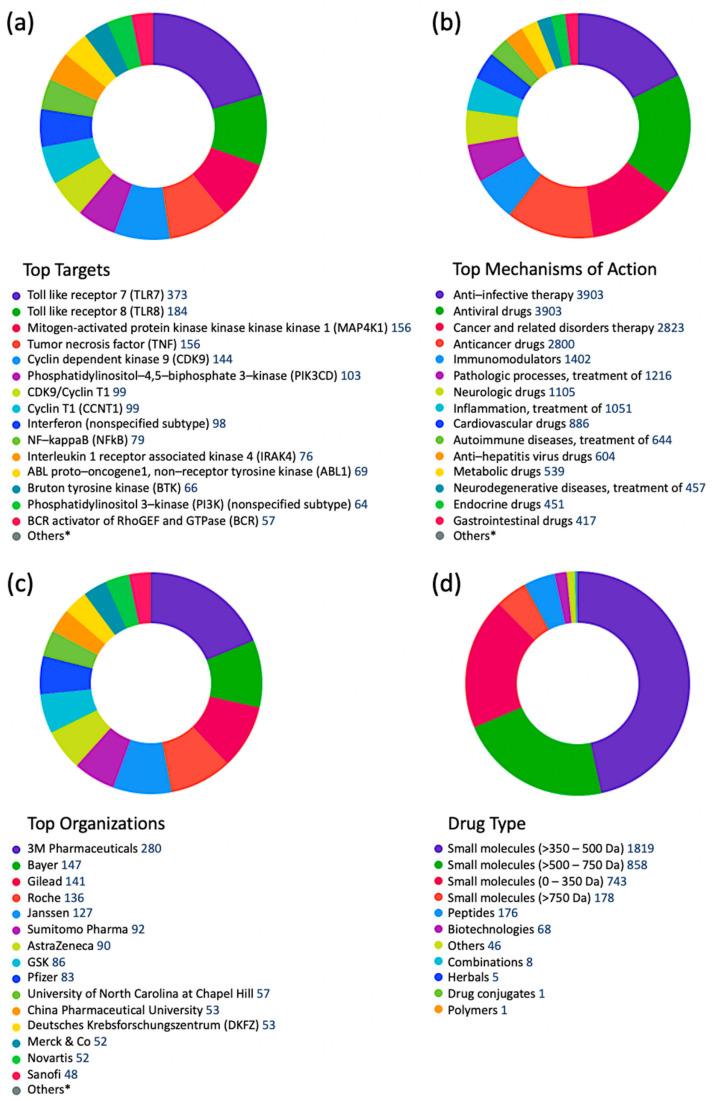
An overview of antiviral drugs modulating signal transduction according to (**a**) top targets, (**b**) top therapeutic groups, (**c**) top organizations, and (**d**) drug types. * Others in (**a**–**c**) indicate that there are other targets, therapeutic groups and mechanisms that are not shown in the figure because they represent a large number of categories with very small percentages. Data source: Cortellis Drug Discovery Intelligence [37], https://www.cortellis.com/drugdiscovery/, accessed on 24 December 2022, ©2022 Clarivate. All rights reserved.

**Figure 4 viruses-15-00568-f004:**
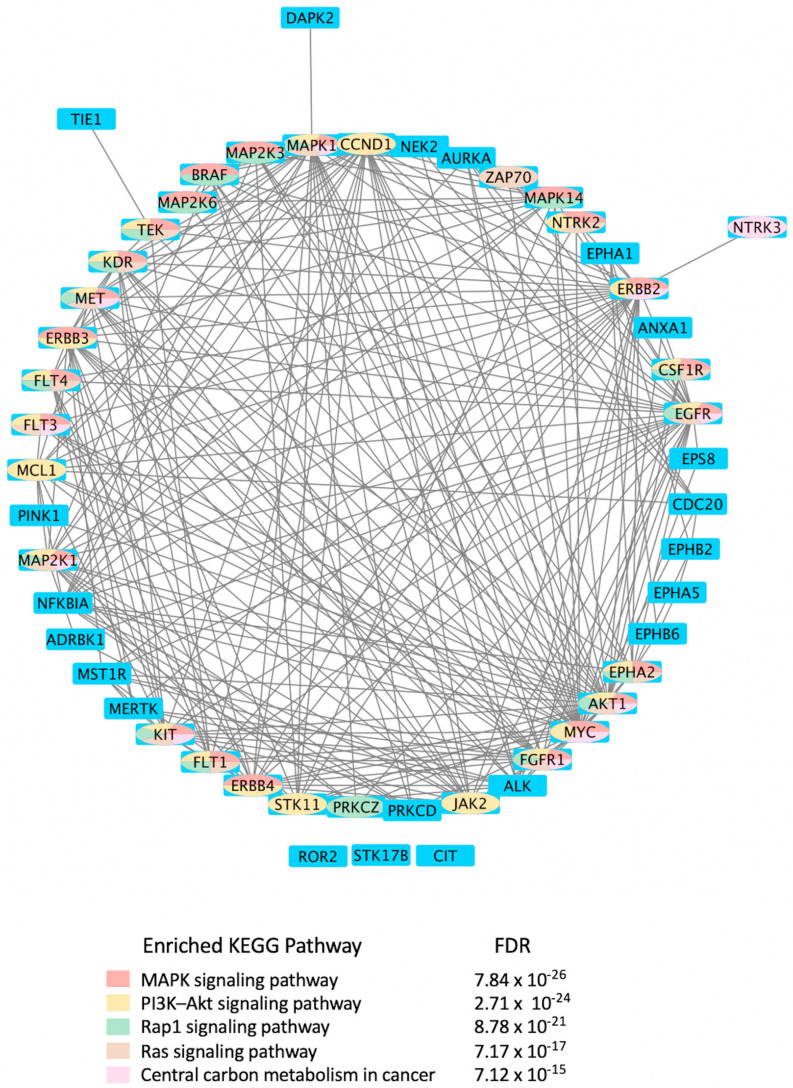
Protein-protein interactions network of kinases explored as diagnostic biomarkers for viral infections. Network nodes are human kinases that are viral disease biomarkers, according to the Cortellis Drug Discovery Intelligence databases [37]. The network was generated using Cytoscape version 3.9.1. Network nodes were colored based on the top five enriched KEGG pathways [73] shown in the color key beneath the network. Blue nodes indicate that the gene/gene product is not part of the top five enriched KEGG pathways shown underneath the network. The pathway prediction false discovery rate (FDR) is reported for each pathway. CDDI [37] was on https://www.cortellis.com/drugdiscovery/, accessed on 26 December 2022, ©2022 Clarivate. All rights reserved.

**Table 1 viruses-15-00568-t001:** Two classification systems for viruses based on the virus genome and Baltimore classes.

Viruses Classification Systems	
Virus Genomes	Baltimore Classes
(1) dsDNA (2) ssDNA (3) ssDNA/dsDNA (4) RT DNA (5) RT RNA (6) dsRNA (7) (+) RNA (8) (−) RNA (9) Others	(1) BCI—dsDNA (2) BCII—ssDNA (3) BCIII—dsRNA (4) BCIV—(+) ssRNA (5) BCV—(−) ssRNA (6) BCVI—RT (+) ssRNA (7) BCVII—RT dsDNA

ds: double-stranded; ss: single-stranded; RT: reverse transcribing; (+): positive-sense; (−): negative-sense; BC: Baltimore Class. Classification information was mined from the literature [13,19,20,21,22].

**Table 2 viruses-15-00568-t002:** A stepwise summary of gene expression and gene replication mechanisms among different classes of viruses [13,22,33,34,35,36].

Virus Class	mRNA and Protein Synthesis	Gene Replication Location and Enzymes
BCI—dsDNA	Direct transcription via hRNApol-II into mRNA; followed by translation dsDNA poxviruses—transcription via viral RNA polymerase to mRNA; followed by translation	Replication inside host cell nuclei, except for poxviruses which replicate inside cytoplasm Host cell DNA replication mechanism enzymes or by viral enzymes, i.e., DNA polymerases
BCII—ssDNA	Replication into dsDNA; followed by transcription via hRNApol-II into mRNA; followed by translation	ssDNA viruses use host enzymes to synthesize dsDNA replication intermediate; followed by ssDNA progeny formation
BCIII—dsRNA	Transcription via RdRp into mRNA; followed by translation	Replication inside cytoplasm replication via RdRp into dsRNA
BCIV—(+) ssRNA	Direct translation	Replication inside cytoplasm Replication via RdRp into (−) ssRNA; followed by replication via RdRp into (+) ssRNA
BCV—(−) ssRNA	Transcription via RdRp into mRNA; followed by translation	Replication inside cytoplasm Replication via RdRp into (+) ssRNA; followed by replication via RdRp into (−) ssRNA
BCVI—RT (+) ssRNA	Reverse transcription via RTz into ssDNA and then into dsDNA; followed by transcription via hRNApol-II into mRNA; followed by translation	Reverse transcription via RTz into ssDNA and then into dsDNA; followed by transcription via hRNApol-II into (+) ssRNA
BCVII—RT dsDNA	Transcription via hRNApol-II into (+) ssRNA; followed by reverse transcription by RTz into ssDNA and then dsDNA; followed by transcription via hRNApol-II into mRNA; followed by translation	Replication inside host cell nuclei Transcription via hRNApol-II into (+) ssRNA; followed by reverse transcription by RTz into ssDNA and then dsDNA

BC: Baltimore Class; ds: double-stranded; ss: single-stranded; RT: reverse transcribing; (+): positive-sense; (−): negative-sense; hRNApol-II: host RNA polymerase II; RdRp: viral RNA-dependent RNA polymerase; RTz: viral reverse transcriptase enzyme.

**Table 4 viruses-15-00568-t004:** Important antiviral kinase inhibitors reported in the biomedical literature.

Compound	Target	Virus	Mechanism of Antivirus
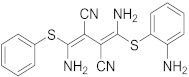 U0126 (**1**) MEK1 IC_50_ = 72 nM MEK2 IC_50_ = 58 nM	MEK 1/2- MAPK	IBV IAV	Inhibits viral RNP release
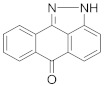 SP600125 (**2**) JNK1/2 IC_50_ = 40 nM JNK3 IC_50_ = 90 nM	JNK1- MAPK	Astrovirus JEV MERS-CoV	Inhibits viral entry Decreases inflammatory cytokines
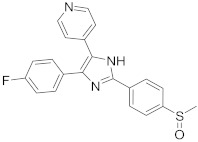 SB203580 (**3**) SAPK2a/p38 IC_50_ = 50 nM	p38-MAPK	MERS-CoV	Inhibits viral entry
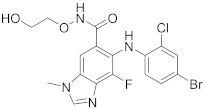 Selumetinib (**4**) MEK1/2 IC_50_ = 14 nM	MEK 1/2- MAPK	MERS-CoV	Antiproliferative activity
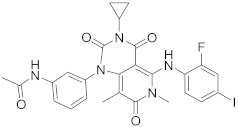 Trametinib (**5**) MEK1/2 IC_50_ = 2 nM	MEK 1/2- MAPK	MERS-CoV	Antiproliferative activity
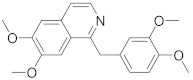 Papaverine (**6**) IC_50 =_ 2.0-36.4 μM	MEK & ERK-MAPK	IAV Paramyxovirus	Inhibits viral RNP release
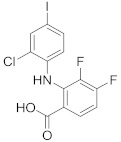 ATR-002 (**7**) MEK1 IC_50_ = 5.73 nM	MEK1-MAPK	IAV IBV	Antiproliferative activity
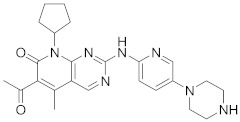 Palbociclib (**8**, PD-0332991) CDK 4 IC_50_ = 11 nM CDK 6 IC_50_ = 16 nM	CDK4/CDK6	HIV-1 HSV-1	CDK4: Antiproliferative activity CDK6: Inhibits reverse transcription
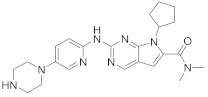 Ribociclib (**9**, LEE011) Breast cancer Novartis, 2017	CDK4/CDK6	SARS-CoV-2	Antiproliferative activity
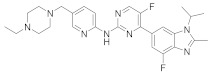 Abemaciclib (**10**, LY2835219) Breast cancer Lilly, 2017 SARS-CoV-2 (IC_50_ = 6.6 mM)	CDK4/CDK6	SARS-CoV-2	Antiproliferative activity
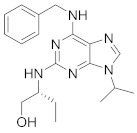 (*R*)-Roscovitine (**11**) CDK2 IC_50_ = 0.7 μM CDK5 IC_50_ = 0.2 μM	CDK2/CDK5	HCMV	Inhibits DNA synthesis
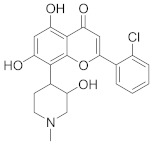 Flavopiridol (**12**) CDK1 IC_50_ = 30 nM CDK2 IC_50_ = 170 nM CDK4 IC_50_ = 100 nM CDK6 IC_50_ = 60 nM	CDK1/CDK2/CDK4	HIV-1 IAV	Antiproliferative activity Synergistic activity with Dinaciclib (**15**)
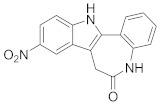 Alsterpaullone (**13**) CDK1 IC_50_ = 35 nM CDK2 IC_50_ = 15 nM	CDK1/CDK2	HIV-1	Antiproliferative activity
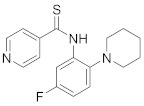 FIT-039 (**14**) CDK9 IC_50_ = 5.8 μM	CDK9	HSV-1 HSV-2 HCMV HAdV-5	Antiproliferative activity Anti-transcription Antiproliferative activity
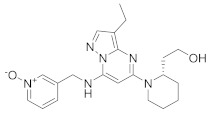 Dinaciclib (**15**) CDK1 IC_50_ = 3 nM	CDK1	SARS-CoV-2	Antiproliferative activity
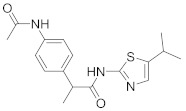 PHA-690509 (**16**) CDK2 IC_50_ = 31 nM	CDK2	ZIKV	Antiproliferative activity
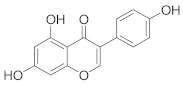 Genistein (**17**) EGFR IC_50_ = 0.6 μM	RTKs	IAV HIV-1 Arenavirus HSV-1	Inhibits viral entry Antiproliferative activity
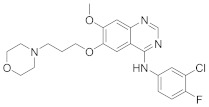 Gefitinib (**18**, Iressa^®^, ZD1839) EGFR IC_50_ = 33 nM GAK K_d_ = 13 nM AAK1 k_d_ > 3000 nM	EGFR	TGEV IAV Rhinovirus	Inhibits viral entry Inhibits infection
FDA- Approved Anticancer Drug, AstraZeneca, 2015			
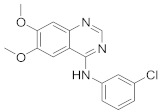 AG1478 (**19**) EGFR IC_50_ = 3 nM	EGFR	TGEV	Inhibits viral intake by IPEC
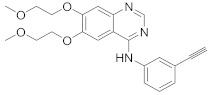 Erlotinib (**20**, Tarceva^®^, CP-358774, R 1415, NSC 718781, OSI 774) EGFR IC_50_ = 2 nM GAK K_d_ = 3.1 nM AAK1 k_d_ = 1.2 μM	EGFR GAK/AAK1	HCV HCV	Inhibits viral entry Inhibits viral entry Inhibits viral assembly
FDA- Approved Anticancer Drug, Genentech, 2004			
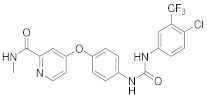 Sorafenib (**21**, Nexavar^®^, BAY43-9006) PDGFR IC_50_ = 20 nM	PDGFR	SARS-CoV-2 DNA virus HBV	Antiproliferative activity
FDA- Approved Anticancer Drug, Onyx, 2005			
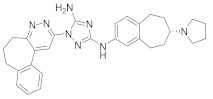 R428 (**23**) AxI IC_50_ = 14 nM	AxI	ZIKV	Inhibits viral entry Activates IFN-1
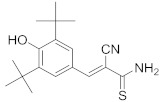 AG879 (**24**) TRK-A IC_50_ = 10 μM	TRK-A	IAV	Antiproliferative activity
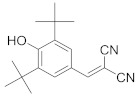 Tyrphostin A9 (**25**) PDGFR IC_50_ = 0.5 μM	PDGFR	IAV	Antiproliferative activity
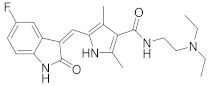 Sunitinib (**26**, SU-11248, Sutent^®^) AAK1 K_d_ = 11 nM GAK K_d_ = 20 nM	AAK1 GAK	HCV	Inhibits viral entry Inhibits viral assembly
FDA- Approved Anticancer Drug, Pfizer, 2006			
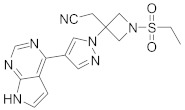 Baricitinib (**27**, LY3009104, Olumiant^®^) AAK1 K_d_ = 8.2 nM GAK K_d_ = 120 nM	AAK1 GAK	SARS-CoV-2	Inhibits viral entry Inhibits viral assembly
FDA- Approved Antirheumatic Agent, Lilly, 2018			
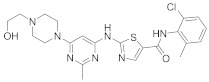 Dasatinib (**28**, BMS-354825, Sprycel^®^) AAK1 K_d_ > 3000 nM GAK K_d_ = 2.6 nM ABL IC_50_ = 2.8 nM	Src Src Fyn Lck ABL	DENV HCV DENV HIV-1 MERS-CoV SARS-CoVMERS-CoV SARS-CoV	Inhibits viral assembly Inhibits viral secretion Potentiates IC_50_ (210 fold) in Huh7.5.1 in combination with Sofosbuvir Antiproliferative Inhibits viral entry Inhibits reverse transcription
FDA- Approved Anticancer Drug, Bristol- Myers Squibb, 2006			
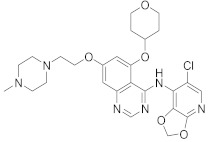 Saracatinib (**29**) Src IC_50_ = 2.7 nM	Src Fyn	DENV DENV MERS-CoV HCoV-229E HCoV-OC43	Inhibits viral assembly Antiproliferative
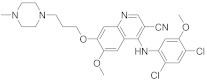 Bosutinib (**30**, SKI-606, Bosulif^®^) Src IC_50_ = 1.2 nM	Src	SARS-CoV-2	Inhibits viral entry
FDA- Approved Anticancer Drug, Pfizer, 2012			
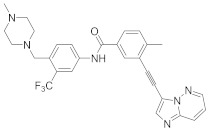 Ponatinib (**31**, AP24534, Iclusig^®^) Src IC_50_ = 5.4 nM	Src	SARS-CoV-2	Inhibits cytokines release
FDA- Approved Anticancer Drug, Ariad, 2012			
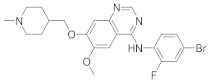 Vandetanib (**32**, ZD6474, Caprelsa^®^) Src IC_50_ = 0.79 nM	Src	SARS-CoV-2 HCoV-229E	Antiproliferative Antiproliferative
FDA- Approved Anticancer Drug, AstraZeneca, 2011			
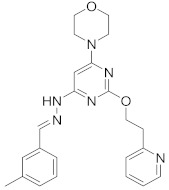 Apilimod (**33**, LAM-002A, STA-5326) Pikfyve IC_50_ = 14 nM	Pikfyve	EBOV SARS-CoV-2	Inhibits viral entry
FDA- Approved Anticancer Drug, AI Therapeutics, Inc.			
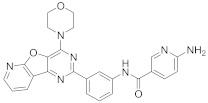 YM-201636 (**34**) Pikfyve IC_50_ = 33 nM	Pikfyve	EBOV	Inhibits viral entry
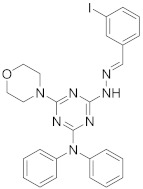 Vacuolin-1 (**35**) Pikfyve Kd = 9 nM	Pikfyve	EBOV	Inhibits viral entry
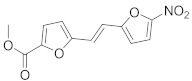 Methyl 5-[2-(5-nitro-2-furyl) vinyl]-2-furoate (**36**) GRK2 IC_50_ = 126 μM	GRK2	IAV	Inhibits viral entry Antiproliferative
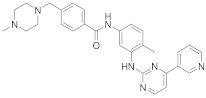 Imatinib (**37**, Gleevec^®^, Glivec^®^) ABL IC_50_ = 220 nM	ABL	MERS-CoV SARS-CoV	Inhibits viral entry
FDA- Approved Anticancer Drug, Novartis, 2001			
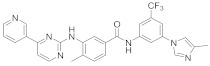 Nilotinib (**38**, AMN107, Tasigna^®^) ABL IC_50_ = 30 nM	ABL	SARS-CoV	Antiproliferative
FDA- Approved Anticancer Drug, Novartis, 2007			
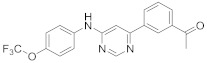 GNF-2 (**39**) ABL IC_50_ = 138 nM	ABL	DENV MERS-CoV SARS-CoV	Inhibits viral entry Antiproliferative Inhibits viral entry
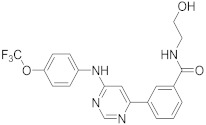 GNF-2 (**40**) ABL IC_50_ = 220 nM	ABL	MERS-CoV SARS-CoV	Inhibits viral entry
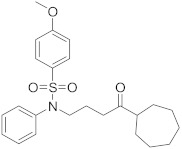 BSA9 (**41**) CaMKII IC_50_ = 0.79 μM	CaMKII	DENV ZIKV	Inhibits viral entry
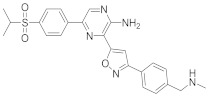 Berzosertib (**42**) ATR K_i_ = 0.2 nM	ATR inhibitor	MERS-CoV SARS-CoV	Antiproliferative
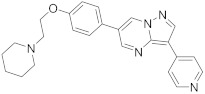 Dorsomorphin (**43**) AMPK K_i_ = 109 nM	AMPK	EBOV SARS-CoV-2	Antiproliferative Antiviral activity
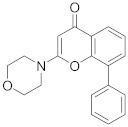 LY294002 (**44**) PI3Kα (IC_50_ = 0.50 μM) PI3Kβ (IC_50_ = 0.97 μM) PI3Kδ (IC_50_ = 0.57 μM)	PI3K	ASFV EBOV HSV-1	Inhibits viral entry
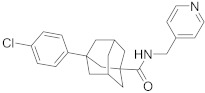 Opaganib (**45**) SphK2 Ki = 9.8 μM	SphK2	SARS-CoV-2	Antiproliferative

**Table 5 viruses-15-00568-t005:** The pros and cons of targeting viral proteins or host proteins.

Drug Target	Pros	Cons
Viral Proteins	Selective approach Viral structural proteins are unique to the pathogen and generally have no human homologs Viral structural proteins tend to form oligomers, leading to a dominant negative effect (i.e., drug does not necessarily need to act on every viral protein subunit to disrupt the oligomer’s function) Good safety profiles in humans Less interference with host genes and proteins, especially those involved in innate immunity.	Binding sites are often difficult to target with small molecules. Prone to drug resistance and cross-resistance due to high viral mutational rates in comparison with human targets. They do not often afford full cures since most of them reduce viral loads below detection limits. Some viruses may hide in host cells and specific organs (e.g., neurons, eyes, and testis).
Host Proteins	Conserved and less prone to drug resistance and cross-resistance. They exhibit broad-spectrum inhibition against many viruses since a specific set of human proteins is commonly hijacked by human viruses to support the viral life cycle.	Lower safety and selectivity profiles in comparison with direct acting antivirals targeting viral proteins. This approach is not suitable for latent viruses. Requires in-depth knowledge of virus-host interactions and their biological significance to viral replication. Need thorough research efforts to elucidate the network biology of innate immunity and consequences of target inhibition/modulation of any of the network genes or proteins.

The information in the table has been summarized based on the information reported in the biomedical literature [331,332,333,334,335].

**Table 6 viruses-15-00568-t006:** Types of kinase inhibitors.

Kinase Inhibitor Type	Characteristics
Type I	Bind to a well-conserved conserved ATP binding pocket that is shared with other kinases and other species. Less selective since they can interact with many kinases. More toxicity issues due to cross-reactivity leading to the clinical failure of initially promising kinase inhibitors.
Type II	They bind to sites adjacent to the ATP binding pocket. Improved selectivity in comparison with type I inhibitors.
Type III-VI	Do not bind to the ATP binding sites; they are not competitive inhibitors to ATP. Greater selectivity profiles Reduced toxicity profiles due to lower off-target effects

## Data Availability

Data supporting the reported results can be requested by contacting the corresponding author directly.

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
