# Peer review of "Targeting Human Proteins for Antiviral Drug Discovery and Repurposing Efforts: A Focus on Protein Kinases"

_viruses, 2023, doi:10.3390/v15020568_

Round 1
Reviewer 1 Report
Reviewer’s Comment
In the review article ‘Targeting Human Proteins for Antiviral Drug Discovery and 2 Repurposing Efforts: A Focus on Protein Kinases’' Hajjo et al. have described direct antivirals in general and described in detail the host factors, especially kinases that are known to involve in the viral life cycle. The authors have done a network analysis of the interaction of kinases.
This review would be beneficial to readers. The manuscript is well written. However, there need to be details of challenges with kinase as antivirals. Following are a few suggestions that might improve the manuscript.
- The authors have included the challenges in targeting host targets broadly (819-845). Kinases interact with many pathways, as indicated by network analysis. The authors may consider including challenges specific to kinases in detail. Also, consider including essential tests to be considered to find out off-target effects.
2. The authors may consider including some of the recent relevant review articles (Eg. PMID: 33539089, PMID: 34081439)
3. Consider including details of reported side effects for the drugs in the Table 3 trials.

Author Response
We thank the reviewer for all the helpful notes and suggestions. We listed our responses below:1) The authors have included the challenges in targeting host targets broadly (819-845). Kinases interact with many pathways, as indicated by network analysis. The authors may consider including challenges specific to kinases in detail. Also, consider including essential tests to be considered to find out off-target effects.
Response: We addressed this request by discussing the challenges specific to kinases in detail, as well as adding the tests to be considered to find out off-target effects. All details can be found in the following text that is highlighted in the revised manuscript under section 10 (10. Pros and Cons of Targeting Host Proteins for Antiviral Drug Discovery) and can be found below:
“In relation to kinases, an in vitro study has investigated the use of specific kinase inhibitors targeting MEK and Src kinases to evaluate their activity against the proliferation of flavivirus infections inside BHK21 and Vero cells [344]. BHK21 and Vero cells are mammalian cells usually used to study the growth of viruses in laboratories [345, 346]. The MEK inhibitors (trametinib and selumetinib) and Src inhibitors (saracatinib and bosutinib), which are being designed to treat cancers, showed antiviral activity against several flaviviruses (Zika virus, dengue virus, and yellow fever virus) [344]. However, the most effective and safest among them was trametinib. Safety was evaluated via the calculation of selectivity index (SI), in which a ratio between the 50% cytotoxic concentration (inhibitors against cells only) and the 50% effective concentration (antiviral activity) is calculated. Trametinib had the highest SI compared to the other kinase inhibitors [344]. Hence, there is a risk of cytotoxicity when repurposing kinase inhibitors to target viral proliferation.
RV replication depends on phosphatidylinositol 4-kinase III beta (PI4KIIIβ) [347]. In an in vitro and in vivostudies, the replication of RV has been prevented by the use of aminothiazole compounds, Compound 1 and Compound 2, which are PI4KIIIβ inhibitors [347]. However, the in vivo study has shown several adverse effects on mice including muscle weakness and difficulty in breathing [347]. Although it was concluded that these adverse effects could be possibly species-specific and not relevant to humans, further in vivo studies are required to check the mechanism of toxicities and their relevance to humans [347].
Kinase inhibitors were also recently being investigated clinically in patients with COVID-19 as extensively summarized by Malekinejad et al. [348]. Clinical trials have studied, with some are still ongoing, the use of JAK/STAT and BTK inhibitors for patients with COVID-19. Although number of JAK/STAT kinase inhibitors, such as NCT05187793 and NCT04390061, resulted in clinical recovery and prevention of severe respiratory failure, respectively, others have shown adverse outcomes [348]. Among the adverse outcomes resulted due to the use of JAK/STAT and BTK inhibitors include hospitalization, the need to mechanical ventilation, extracorporeal membrane oxygenation (ECMO), respiratory failure, the need of supplemental oxygen, renal failure, disease progression, ICU admission, and death [348].”
“Targeted therapies that involve the use of enzyme inhibitors, such as kinase inhibitors, can alter normal physiological cell functions, and thus altering normal cell behavioral phenotype, via interacting with other pathways that are not intended to act upon [359-361]. These abnormal effects could arise either directly due to off-target interactions with additional proteins within the targeted pathway [362, 363] or indirectly due to crosstalk between the targeted pathway and other pathways regulating a behavioral response [364]. In addition, off-target effects could arise due to retroactivity; in which a downstream disruption in a signaling pathway produces an upstream effect even though there is no association of negative feedback inhibition [365]. Although retroactivity do occur naturally in covalently modified cascades, the movement of signals within signaling pathways is usually in a downstream manner [366]. Kinase inhibitors have shown to produce off-target effects via retroactivity [366]. Cell behaviors and off-target effects in the presence of kinase inhibitors or other drugs could be predicted computationally using partial least square regression framework based on the signals of several key signaling pathways [367-369].”
2) The authors may consider including some of the recent relevant review articles (Eg. PMID: 33539089, PMID: 34081439).
Response: We referenced these two articles under section 9 (9. Repurposing Old Kinase Inhibitors as Antivirals). They are now references 337 and 338 in the reference list. Both are highlighted in yellow in the revised manuscript.
3) Consider including details of reported side effects for the drugs in the Table 3 trials.
Response: We added a new column to Table 3 to cover details of reported side effects. The newly added information in Table 3 is highlighted in yellow.
Reviewer 2 Report
Manuscript viruses-2185445 by Hajjo et al. describes the function of host kinases, the role of these kinases in virus life cycle as potential target for antiviral development. The manuscript also describes the status quo of antiviral discovery and development pipelines including those targeting host kinases. Lastly, the manuscript describes the potency, efficacy, mechanisms, and side effects of kinase inhibitors in in vitro and in vivo studies.
The manuscript is a good review and contains substantial amount of information for the field, to include contents describing selected host kinases involved in multiple stages of viral proliferation - virus attachment, entry, genome replication, assembly and release, and summarizing antiviral activities targeting these signal transduction pathways, which could open a new window for drug discovery. Another interesting content for me is the repurposing of drugs in the fields of oncology and functional dysregulation (e.g. selumetinib, papaverine, et. al.) to the field of virology, which would be critical to extend functions of existing drugs and save time for drug discovery.
I only have minor comments:
In line 243, and lines 254 and 255, what is the difference between the two search criteria needs more clarity.
In Figure 4, what does color blue code for?
In session 7 'The Most Frequently Targeted Kinases for Antiviral Drug Development', to improve clarity and consistency, it's recommended to combine several sub-sessions belongs to Receptor Tyrosine Kinases together (e.g. EGFR, AXL RTK, JAK-STAT pathways and MER), and to combine CCNT1 cyclin T1 with CDKs. Not sure, if really need CCNT1 cyclin T1 sub-session since it's not targeting kinase but protein binds to CDK.
Author Response
We thank the reviewer for the helpful comments and suggestions. Here is a point-by-point response:
1) In line 243, and lines 254 and 255, what is the difference between the two search criteria needs more clarity.
Response: We added clarifying statements after listing the search criteria. Added text is highlighted in yellow in the revised manuscript (and indicated by the writing in italics below).
These search criteria on L243, allow the retrieval of all drugs and biologics that have been linked to antiviral activity and can also modulate a kinase. The search described on L254-255 will retrieve antivirals being developed to combat viral diseases. This clarifications has been added to the manuscript and highlighted in yellow:
“To get a better idea about which kinases have any potential for antiviral drug discovery, we mined all drugs and biologics in CDDI database using the following search criteria: “condition = viral infection” and “mechanism = kinase”. These search criteria allow the retrieval of all drugs and biologics that have been linked to antiviral activity and can also modulate a kinase. Our search resulted in 1251 drugs and biologics, of which, 1204 drugs are still in the biological testing phase and have not progressed in the drug development pipeline yet.”
“A more targeted search of the CDDI database was performed using the developmental status condition as follows: “development status condition = infection, viral” and “mechanism of action = drugs targeting kinases”. Imposing these filtering criteria ensured that the retrieved drugs and biologics have antiviral effects and are being developed precisely to combat viral infections by targeting kinases. This search resulted in seven drugs and biologics (Table 3). The drug targets of these drugs include casein kinase 2 (CK2), MAP2K, MAPK, MAPK p38, and CDK1.”
2) In Figure 4, what does color blue code for?
Response: When the node is blue it indicates that the node is not part of the five listed pathways. We added this clarification to the figure legend.
Added clarifying text is highlighted in yellow in the revised manuscript (and indicated by the writing in italics below).
“Figure 4. Protein-protein interactions network of kinases explored as diagnostic biomarkers for viral infections. Network nodes are human kinases that are viral disease biomarkers according to the Cortellis Drug Discovery Intelligence databases [37]. The network was generated using Cytoscape version 3.9.1. Network nodes were colored based on top five enriched KEGG pathways [65] shown in the color key beneath the network. Blue nodes indicate that the gene/gene product is not part of top five enriched KEGG pathways shown underneath the network. The pathway prediction false discovery rate (FDR) is reported for ea ch pathway. CDDI [37] was accessed on 26 December 2022, https://www.cortellis.com/drugdiscovery/ ©2022 Clarivate. All rights reserved.”
3) In session 7 'The Most Frequently Targeted Kinases for Antiviral Drug Development', to improve clarity and consistency, it's recommended to combine several sub-sessions belongs to Receptor Tyrosine Kinases together (e.g. EGFR, AXL RTK, JAK-STAT pathways and MER), and to combine CCNT1 cyclin T1 with CDKs. Not sure, if really need CCNT1 cyclin T1 sub-session since it's not targeting kinase but protein binds to CDK.
Response: We thank the reviewer for this suggestion. We revised section 7 according to the reviewer’s comments. All changes are are highlighted in yellow in the revised manuscript.